# CONSISTENCY MODELS MADE EASY

**Zhengyang Geng**[1]  **Ashwini Pokle**[1]  **Weijian Luo**[2]  **Justin Lin**[1]  **J. Zico Kolter**[1]

[1]CMU  [2]Peking University

## ABSTRACT

Consistency models (CMs) offer faster sampling than traditional diffusion models, but their training is resource-intensive. For example, as of 2024, training a state-of-the-art CM on CIFAR-10 takes one week on 8 GPUs. In this work, we identify the "curse of consistency" for training such models and propose an effective training scheme that largely mitigates this issue and improves the efficiency of building such models. Specifically, by expressing CM trajectories via the differential equation, we argue that diffusion models can be viewed as a special case of CMs. We can thus fine-tune a consistency model starting from a pretrained diffusion model and progressively approximate the full consistency condition to stronger degrees over the training process. Our resulting method, which we term Easy Consistency Tuning (ECT), achieves vastly reduced training times while improving upon the quality of previous methods: for example, ECT achieves a 2-step FID of 2.73 on CIFAR10 within 1 hour on a single A100 GPU, matching Consistency Distillation trained for hundreds of GPU hours. Owing to this computational efficiency, we investigate the scaling laws of CMs under ECT, showing that they obey the classic power law scaling, hinting at their ability to improve efficiency and performance at larger scales. Our code is available.

## 1 INTRODUCTION

Diffusion Models (DMs) (Ho et al., 2020; Song et al., 2021a), or Score-based Generative Models (SGMs) (Song et al., 2020; 2021b), have vastly changed the landscape of visual content generation with applications in images (Rombach et al., 2021; Saharia et al., 2022; Ho et al., 2022a; Dhariwal and Nichol, 2021; Hatamizadeh et al., 2023; Ramesh et al., 2021), videos (Brooks et al., 2024; Blattmann et al., 2023; Bar-Tal et al., 2024; Ho et al., 2022b; Gupta et al., 2023), and 3D objects (Poole et al., 2022; Wang et al., 2024a; Lee et al., 2024; Chen et al., 2024; Babu et al., 2023). DMs progressively transform a data distribution to a known prior distribution (e.g. Gaussian noise) according to a stochastic differential equation (SDE) (Song et al., 2021b) and train a model to denoise noisy observations. Samples can be generated via a reverse-time SDE that starts from noise and uses the trained model to progressively denoise it. However, sampling from a DM naively requires hundreds to thousands of model evaluations due to the curvature of the diffusion sampling trajectory (Karras et al., 2022), making the entire generative process slow. Many approaches have been proposed to address this issue, including training-based techniques such as distillation (Luhman and Luhman, 2021; Salimans and Ho, 2022; Luo et al., 2024; Gu et al., 2023; Sauer et al., 2023; Geng et al., 2024; Yin et al., 2023; Nguyen and Tran, 2023), adaptive compute architectures for the backbone model (Moon et al., 2023; Tang et al., 2023), as well as training-free methods such as fast samplers (Kong and Ping, 2021; Lu et al., 2022a; Zhang and Chen, 2022; Zhou et al., 2023; Xue et al., 2024) or interleaving small and large backbone models during sampling (Pan et al., 2024). However, sample quality and mode coverage achieved by these speedup techniques still struggles when the number of model evaluations is reduced to below 5.

Consistency Models (CMs) (Song et al., 2023) are a new family of generative models, closely related to diffusion models, that have demonstrated promising results as faster generative models. These models learn a mapping between noise and data, and all the points of the sampling trajectory map to the same initial data point. Owing to this condition, consistency models are capable of generating high-quality samples in 1-2 model evaluations. The best such models so far, built using improved Consistency Training (iCT) (Song and Dhariwal, 2023), have pushed the quality of images generated

by 1-step CMs trained from scratch to a level comparable with SoTA DMs using thousands of steps for sampling. Unfortunately, CMs remain time-consuming and practically challenging to train: the best practice takes many times longer than similar-quality DMs while involving complex hyperparameter choices in the training process. In total, this has substantially limited the uptake of CMs within the community.

In this work, we introduce a differential perspective on consistency models, leading to the formulation of the differential consistency condition in continuous time. This insight reveals the link between diffusion models and consistency models, viewing DMs as a special case of CMs with loose discretization. This observation motivates us to smoothly interpolate from DM to CM by progressively tightening the consistency condition, bootstrapping pretrained DMs to 1-step CMs w/o using extra frozen teachers. We term this strategy as *Easy Consistency Tuning (ECT)*, which includes diffusion pretraining as a special stage of the continuous time training schedule.

ECT significantly improves both training efficiency and performance. On ImageNet 64×64 (Deng et al., 2009), ECT achieves superior 1-step and 2-step sample quality compared to the prior art. Similarly, on CIFAR-10 (Krizhevsky, 2009) 2-step sample quality of ECT surpasses previous methods. The total cost of the pretraining and tuning scheme requires only $1/4 \sim 1/3$ of the computational resources (FLOPs) used by the current state-of-the-art method, iCT (Song and Dhariwal, 2023), while the tuning stage can be remarkably lightweight, typically accounting for $10\%$ or less of the overall cost and further benefiting from scaling.

Leveraging ECT's computational efficiency, we conduct the first study into the scaling behaviors of CMs, revealing the classic power law scaling for model size, FLOPs, and training compute. The scaling also suggests a sweet spot of using smaller few-step CMs over larger 1-step CMs in certain scenarios. This computational efficiency enables us to explore the design space of CMs by using the tuning stage as a proxy. Notably, tuning findings, such as weighting functions, can improve the pretraining stage and, in turn, enhance the overall pretraining + tuning pipeline for CMs.

In short, we summarize our contributions as follows:

- We develop Easy Consistency Tuning (ECT), a pretraining + tuning scheme for training CMs in continuous time, demonstrating significant efficiency and performance gains compared to the current best practices for training CMs.
- We investigate CMs' scaling behaviors for the first time and reveal the classic power law.
- We explore the design space of CMs through ECT, introducing the continuous-time schedule and better weighting functions for CMs.

## 2 PRELIMINARIES

**Diffusion Models.** Let $p_{\text{data}}(\mathbf{x}_0)$ denote the data distribution. Diffusion models (DMs) perturb this distribution by adding monotonically increasing i.i.d. Gaussian noise with standard deviation $\sigma(t)$ from $t = 0$ to $T$ such that $p_t(\mathbf{x}_t|\mathbf{x}_0) = \mathcal{N}(\mathbf{x}_0, \sigma^2(t)\boldsymbol{I})$, and $\sigma(t)$ is chosen such that $\sigma(0) = \sigma_{\min}$ and $\sigma(T) = \sigma_{\max}$. This process is described by the following SDE (Song et al., 2021b)

$$\mathrm{d}\mathbf{x} = g(t)\mathrm{d}\mathbf{w}, \tag{1}$$

where $\mathbf{w}$ is the standard Wiener process, and $g(\cdot) : \mathbb{R} \to \mathbb{R}$ is the diffusion coefficient. Samples can be generated by solving the reverse-time SDE starting from $t = T$ to $0$ and sampling $\mathbf{x}_T \sim \mathcal{N}(0, \sigma_{\max}^2\boldsymbol{I})$. (Song et al., 2021b) show that this SDE has a corresponding ODE, called the probability flow ODE (PF-ODE), whose trajectories share the same marginal probability densities as the SDE. We follow the notation in (Karras et al., 2022) to describe the ODE as

$$\mathrm{d}\mathbf{x}_t = -\dot{\sigma}(t)\sigma(t)\nabla_{\mathbf{x}_t}\log p_t(\mathbf{x}_t)\mathrm{d}t, \tag{2}$$

where $\nabla_{\mathbf{x}_t}\log p_t(\mathbf{x}_t)$ denotes the score function. Prior works (Karras et al., 2022; Song et al., 2023) set $\sigma(t) = t$ which yields

$$\frac{\mathrm{d}\mathbf{x}_t}{\mathrm{d}t} = -t\nabla_{\mathbf{x}_t}\log p_t(\mathbf{x}_t) \tag{3}$$

We will follow this parameterization in the rest of this paper. Note that time is the same as noise level with this parametrization, and we will use these two terms interchangeably.

**Consistency Models.** CMs are built upon the PF-ODE in Eq. (3), which establishes a bijective mapping between data distribution and noise distribution. CMs learn a *consistency function* $f(\mathbf{x}_t, t)$ that maps the noisy image $\mathbf{x}_t$ back to the clean image $\mathbf{x}_0$

$$f(\mathbf{x}_t, t) = \mathbf{x}_0. \tag{4}$$

By taking the time derivative of both sides, given by continuous-time consistency models (Song et al., 2023).

$$\frac{\mathrm{d}f}{\mathrm{d}t} = 0. \tag{5}$$

However, this differential form $\frac{\mathrm{d}f}{\mathrm{d}t} = 0$ alone is not sufficient to guarantee that the model output will match the clean image, as there exist trivial solutions where the model maps all the inputs to a constant value, such as $f(\mathbf{x}_t, t) \equiv 0$. To eliminate these collapsed solutions, (Song et al., 2023; Song and Dhariwal, 2023) impose the boundary condition for $f(\mathbf{x}_t, t) = \mathbf{x}_0$ via model parameterization:

$$\boxed{f(\mathbf{x}_t, t) = \mathbf{x}_0 \Leftrightarrow \frac{\mathrm{d}f}{\mathrm{d}t} = 0, f(\mathbf{x}_0, 0) = \mathbf{x}_0.} \tag{6}$$

This boundary condition $f(\mathbf{x}_0, 0) = \mathbf{x}_0$ ensures that the model output matches the clean image when the noise level is zero. Together, the differential form in Eq. (5) and the boundary condition define the ***consistency condition***.

Prior works (Karras et al., 2022; Song et al., 2023; Song and Dhariwal, 2023) impose this boundary condition by parametrizing the CM as

$$f_\theta(\mathbf{x}_t, t) = c_{\text{skip}}(t)\,\mathbf{x}_t + c_{\text{out}}(t)\,F_\theta(\mathbf{x}_t, t), \tag{7}$$

where $\theta$ is the model parameter, $F_\theta$ is the network to train, and $c_{\text{skip}}(t)$ and $c_{\text{out}}(t)$ are time-dependent scaling factors such that $c_{\text{skip}}(0) = 1$, $c_{\text{out}}(0) = 0$. This parameterization guarantees the boundary condition by design. We discuss specific choices of $c_{\text{skip}}(t)$ and $c_{\text{out}}(t)$ in Appendix D.

**Training Techniques for Consistency Models.** During training, CMs first discretize the PF-ODE into $N - 1$ subintervals with boundaries given by $t_{\min} = t_1 < t_2 < \ldots < t_N = T$. The model is trained on the following CM loss, which minimizes a metric between adjacent points on the sampling trajectory

$$\arg\min_\theta \mathbb{E}\left[ w(t_i)\mathbf{d}(f_\theta(\mathbf{x}_{t_{i+1}}, t_{i+1}), f_{\theta^-}(\tilde{\mathbf{x}}_{t_i}, t_i)) \right]. \tag{8}$$

Here, $\mathbf{d}(\cdot, \cdot)$ is a metric function, the $f_\theta$ indicates the consistency function, $f_{\theta^-}$ indicates an exponential moving average (EMA) of the past values of $f_\theta$, and $\tilde{\mathbf{x}}_{t_i} = \mathbf{x}_{t_{i+1}} - (t_i - t_{i+1})t_{i+1}\nabla_{\mathbf{x}_{t_{i+1}}} \log p_{t_{i+1}}(\mathbf{x}_{t_{i+1}})$. Further, the discretization curriculum $N$ should be adjusted during training to achieve strong performance.

In the seminal work, Song et al. (2023) use Learned Perceptual Similarity Score (LPIPS) (Zhang et al., 2018) as a metric function, set $w(t_i) = 1$ for all $t_i$, and sample $t_i$ according to the sampling scheduler by Karras et al. (2022): $t_i = \left( t_{\max}^{1/\rho} + \frac{i}{N-1}(t_{\min}^{1/\rho} - t_{\max}^{1/\rho}) \right)^\rho$ for $i \in \mathcal{U}[1, N-1]$ and $\rho = 7$. Further, the score function $\nabla_{\mathbf{x}_t} \log p(\mathbf{x}_t)$ can either be estimated from a pretrained diffusion model, which results in Consistency Distillation (CD), or can be estimated with an unbiased score estimator in Eq. (9), corresponding to consistency training (CT).

$$\nabla_{\mathbf{x}_t} \log p(\mathbf{x}_t) = \mathbb{E}\left[ \nabla_{\mathbf{x}_t} \log p(\mathbf{x}_t|\mathbf{x}_0) \Big| \mathbf{x}_t \right] = \mathbb{E}\left[ -\frac{\mathbf{x}_t - \mathbf{x}_0}{t^2} \Big| \mathbf{x}_t \right], \tag{9}$$

The follow-up work, iCT (Song and Dhariwal, 2023), introduces techniques that significantly improve the performance of CMs. First, the LPIPS metric, which introduces undesirable bias in generative modeling, is replaced with a Pseudo-Huber metric. Second, the network $f_{\theta^-}$ does not maintain an EMA of the past values of $f_\theta$. Third, iCT replaces the uniform weighting scheme $w(t_i) = 1$ with $w(t_i) = \frac{1}{t_{i+1} - t_i}$. Further, the scaling factors of noise embeddings and dropout are carefully selected. Fourth, iCT introduces a discretization curriculum during training:

$$N(m) = \min(s_0 2^{\lfloor \frac{m}{M'} \rfloor}, s_1) + 1, \quad M' = \left\lfloor \frac{M}{\log_2 \lfloor \frac{s_1}{s_0} \rfloor + 1} \right\rfloor, \tag{10}$$

where $m$ is the current number of iterations, $M$ is the total number of iterations, $\sigma_{\max}$ and $\sigma_{\min}$ is the largest and smallest noise level for training, $s_0 = 10$ and $s_1 = 1280$ are hyperparameters. Finally, during training, iCT samples $i \sim p(i) \propto \mathrm{erf}\left(\frac{\log(t_{i+1}) - P_{\mathrm{mean}}}{\sqrt{2}P_{\mathrm{std}}}\right) - \mathrm{erf}\left(\frac{\log(t_i) - P_{\mathrm{mean}}}{\sqrt{2}P_{\mathrm{std}}}\right)$ from a discrete Lognormal distribution, where $P_{\mathrm{mean}} = -1.1$ and $P_{\mathrm{std}} = 2.0$.

## 3 PROBING CONSISTENCY MODELS

We first discuss the finite difference approximation on continuous-time consistency models. Next, we will analyze this loss objective and highlight the challenges of training CMs with it. Based on this analysis, we present our method, Easy Consistency Tuning (ECT). ECT is a simple, principled approach to efficiently train CMs to meet the consistency condition. The resulting CMs can generate high-quality samples in 1 or 2 sampling steps.

### 3.1 THE "CURSE OF CONSISTENCY" AND ITS IMPLICATIONS

To learn the consistency condition, continuous-time consistency models (Song et al., 2023) could be discretized using finite-difference approximation:

$$0 = \frac{\mathrm{d}f}{\mathrm{d}t} \approx \frac{f_\theta(\mathbf{x}_t) - f_\theta(\mathbf{x}_r)}{t - r} \tag{11}$$

where $\mathrm{d}t \approx \Delta t = t - r$, $t > r >= 0$, and $f_\theta(\mathbf{x}_t)$ denotes $f_\theta(\mathbf{x}_t, t)$. For a given clean image $\mathbf{x}_0$, we produce two perturbed images $\mathbf{x}_t$ and $\mathbf{x}_r$ using *shared noise direction* $\boldsymbol{\epsilon} \sim p(\boldsymbol{\epsilon})$ at two noise levels $t$ and $r$, i.e., $\mathbf{x}_t = \mathbf{x}_0 + t \cdot \boldsymbol{\epsilon}$ and $\mathbf{x}_r = \mathbf{x}_0 + r \cdot \boldsymbol{\epsilon}$.

Given the discretization, the objective function for CMs in Eq. (8) can be written as

$$\arg \min_\theta \mathbb{E}_{\mathbf{x}_0, \boldsymbol{\epsilon}, t} \left[ w(t, r) \mathbf{d}(f_\theta(\mathbf{x}_t), f_{\mathrm{sg}(\theta)}(\mathbf{x}_r)) \right], \tag{12}$$

while removing the discrete training schedules. The consistency condition in Eq. (6) holds by eliminating discretization errors, *i.e.,* $\Delta t = (t - r) \to 0$.

However, the consistency loss can be challenging to optimize when $\Delta t \to 0$. This is because the prediction errors from each discretization interval accumulate, leading to slow training convergence or, in the worst case, divergence. To further elaborate, consider a large noise level $T$. We first split the interval from zero to maximum noise level $T$ into $N$ smaller consecutive subintervals.

$$\|f_\theta(\mathbf{x}_T) - \mathbf{x}_0\| \le \sum_i^N \|f_\theta(\mathbf{x}_{t_i}) - f_\theta(\mathbf{x}_{r_i})\| \le Ne_{\max}, \tag{13}$$

where $r_1 = 0 < t_1 = r_2 < \cdots < t_{N-2} = r_N < t_N = T$, and $e_{\max} = \max_i \|f_\theta(\mathbf{x}_{t_i}) - f_\theta(\mathbf{x}_{r_i})\|$, for $i = 1, \cdots, N$. Ideally, we want both $N$ and $e_{\max}$ to be small so that this upper bound is small. But in practice, there is a trade-off between these two terms. As $\Delta t_i = (t_i - r_i) \to 0$, $e_{\max}$ decreases because $\mathbf{x}_{t_i}$ and $\mathbf{x}_{r_i}$ will be close, and it is easier to predict both $f_\theta(\mathbf{x}_{r_i})$ and $f_\theta(\mathbf{x}_{t_i})$. However,

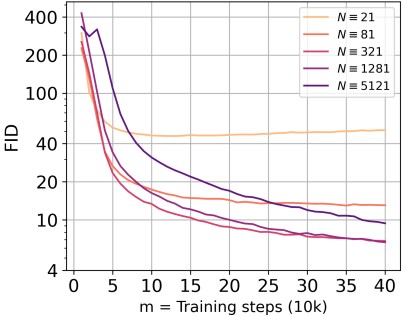

Figure 1: The "Curse of Consistency": The consistency condition holds at $\Delta t = \mathrm{d}t$. However, the training dynamics converges more slowly and is less stable as $\Delta t \to 0$ (*i.e., $N \to \infty$*).

$N$ will increase as $\Delta t_i \to 0$. In contrast, for a large $\Delta t_i$, vice-versa holds true. It is difficult to theoretically estimate the rate at which $e_{\max}$ decreases when $\Delta t_i = t_i - r_i \to 0$, as it depends on the optimization process—specifically, how effectively we train the model to minimize the consistency error in each interval. If $e_{\max}$ decreases more slowly than $N$, the product of the two can increase instead, resulting in a worse prediction error $\|f_\theta(\mathbf{x}_T) - \mathbf{x}_0\|$.

The insight from the above observation is that when training from scratch by strictly following the differential form $\mathrm{d}f/\mathrm{d}t = 0$ with a tiny $\Delta t_i \approx 0$, the resulting model might converge slowly due to accumulated consistency errors from each interval $\Delta t_i$. To investigate this hypothesis, we conducted experiments training a series of CMs using different fixed numbers of intervals $N$, and corresponding

$\Delta t$ values for the consistency condition, as illustrated in Fig. 1. Our observations align with Song and Dhariwal (2023), with the key distinction that we use a fixed $N$ for this analysis, as this approach provides better insight into how much the precise approximation of the consistency condition matters and isolates the effect of discretization errors.

## 3.2 EASY CONSISTENCY TUNING (ECT)

The discussion in Sec. 3.1 highlighted the training instability issue that can arise when we naively optimize for the differential consistency condition with the loss objective in Eq. (12) while directly following $\Delta t \approx 0$. In this section, we propose several strategies that largely alleviate the aforementioned issues and improve the efficiency of CMs. We term this approach Easy Consistency Tuning (ECT), as it effectively balances training stability and model performance while simplifying the CM training process. ECT follows a two-stage approach: diffusion pretraining, followed by consistency tuning, which we will detail in the following subsections with a summary in Alg. 1.

**Diffusion Pretraining + Consistency Tuning.**   Drawing inspiration from iCT's adaptive discrete-time schedule (Song and Dhariwal, 2023), we start ECT with a large $\Delta t$, and gradually shrink $\Delta t \to 0$. In our problem setup, since $t > r \geq 0$, we have the largest possible $\Delta t = t$ with $r = 0$, which yields

$$\arg\min_{\theta} \|f_\theta(\mathbf{x}_t) - f_{\mathrm{sg}(\theta)}(\mathbf{x}_r)\| = \|f_\theta(\mathbf{x}_t) - f_{\mathrm{sg}(\theta)}(\mathbf{x}_0)\| = \|f_\theta(\mathbf{x}_t) - \mathbf{x}_0\|. \tag{14}$$

Training a model with this loss is *identical* to diffusion model/Score SDE (Ho et al., 2020; Song et al., 2021b). This observation suggests a learning scheme that smoothly interpolates from DMs $\Delta t = t$ to CMs $\Delta t = \mathrm{d}t$ by gradually shrinking $\Delta t \to 0$ during training by gradually tightening the consistency condition. With this reasoning, diffusion pretraining can be considered as a special case of consistency training with a loose discretization of the consistency condition. Therefore, in practice, we start ECT with a pretrained diffusion model, resulting in a training scheme of pretraining+tuning. Another benefit of this initialization is that during training, especially in the initial stages, it ensures good targets $f_{\mathrm{sg}(\theta)}(\mathbf{x}_r)$ in the loss objective, avoiding trivial solutions.

We highlight two advantages of this learning scheme: 1. The pretraining+tuning scheme of ECT outperforms iCT's training-from-scratch approach with lower overall computational cost (see Sec. 4.1). 2. Tuning could serve as an efficient proxy for exploring the CM design space. Given a pretrained diffusion model, insights gained during tuning can be applied to improve the pretraining stage, resulting in a more refined overall CM training pipeline. Please refer to Appendix B for details.

**Continuous-time Training Schedule.**   We investigate the design principles of a continuous-time schedule whose "boundary" condition yields standard diffusion pretraining, *i.e.,* constructing training pairs of $r = 0$ for all $t$ at the beginning. Note that this is unlike the discrete-time schedule used in iCT (See Sec. 2 and Appendix A). We consider overlapped intervals for consistency models, which allows for factoring $p(t, r) = p(t)\, p(r|t)$ and continuous sampling of infinite $t$ from noise distribution $p(t)$, for instance, $\mathrm{LogNormal}(P_{\mathrm{mean}}, P_{\mathrm{std}})$, and $r \sim p(r|t)$.

Since we need to shrink $\Delta t \to 0$ as the training progresses, we augment the $p(r|t)$ to depend on training iterations, $p(r|t, \mathrm{iters})$, to control $\Delta t = (t - r) \to 0$. We parametrize $p(r|t, \mathrm{iters})$ as

$$\frac{r}{t} = 1 - \frac{1}{q^a}n(t) = 1 - \frac{1}{q^{\lfloor \mathrm{iters}/d \rfloor}}n(t), \tag{15}$$

where we take $n(t) = 1 + k\,\sigma(-bt) = 1 + \frac{k}{1+e^{bt}}$ with $\sigma(\cdot)$ as the sigmoid function, iters refers to training iterations. In general, we set $q > 1$, $k = 8$ and $b = 1$. Since $r \geq 0$, we also clamp $r$ to satisfy this constraint. At the beginning of training, this mapping function produces $r/t = 0$, which recovers the diffusion pretraining. We discuss design choices of this function in Appendix A.

**Weighting function.**   Weighting functions usually lead to a substantial difference in performance in DMs, and the same holds for CMs. When considering iCT's weighting function, $w(t, r) = 1/t-r$, derived from the finite difference approximation, it couples the timestep weighting with $p(r|t)$. Instead, we consider more flexible timestep weighting independent from the finite difference approximation. We formulate the weighting function as

$$w(t) = \bar{w}(t) \cdot w(\Delta) = \bar{w}(t) \cdot \frac{1}{(\|\Delta\|_2^2 + c^2)^p}, \tag{16}$$

---

**Algorithm 1** Easy Consistency Tuning (ECT)

---

**Input:** Dataset $\mathcal{D}$, a pretrained diffusion model $\phi$, mapping function $p(r \mid t, \text{Iters})$, weighting function $w(t)$.
**Init:** $\theta \leftarrow \phi$, $\text{Iters} = 0$.
**repeat**
    Sample $\mathbf{x}_0 \sim \mathcal{D}$, $\boldsymbol{\epsilon} \sim p(\boldsymbol{\epsilon})$, $t \sim p(t)$, $r \sim p(r \mid t, \text{Iters})$
    Compute $\mathbf{x}_t = \mathbf{x}_0 + t \cdot \boldsymbol{\epsilon}$, $\mathbf{x}_r = \mathbf{x}_0 + r \cdot \boldsymbol{\epsilon}$, $\Delta t = t - r$
    $L(\theta) = w(t) \cdot \mathbf{d}(f_\theta(\mathbf{x}_t), f_{\text{sg}(\theta)}(\mathbf{x}_r))$               ▷ sg is stop-gradient operator
    $\theta \leftarrow \theta - \eta \nabla_\theta L(\theta)$
    $\text{Iters} = \text{Iters} + 1$
**until** $\Delta t \to 0$ **return** $\theta$                                    ▷ ECM

---

where $\Delta = f(\mathbf{x}_t) - f(\mathbf{x}_r)$. We define $w(\Delta)$ as adaptive weighting since it operates on a per data-sample basis and adapts to different magnitudes of error. In particular, $p = 1/2$ corresponds to the Pseudo-Huber loss introduced in the prior art (See Appendix A for more details).

In general, we notice that CMs' generative capability greatly benefits from weighting functions that control the variance of the gradients across different noise levels. We direct the reader to Appendix B for a detailed overview of various choices of $\bar{w}(t)$ and $w(\Delta)$ considered in this work.

**Dropout.** In line with (Song and Dhariwal, 2023), we find that CMs benefit significantly from dropout (Hinton et al., 2012). On ImageNet 64×64, we note that ECT benefits from a surprisingly high dropout rate. When increasing the dropout rate from 0.10 to 0.40, the 2-step FID decreases from 4.53 to 3.24. Finally, we note that the dropout rate tuned at a given weighting function $w(t)$ transfers well to the other weighting functions, thereby reducing the overall cost of hyperparameter tuning.

## 4 EXPERIMENTS

This section compares different learning schemes and investigates scaling laws of ECT, while more experiments on design choices and scaling are shown in Appendix B. We evaluate the efficiency and scalability of ECT on two datasets: CIFAR-10 (Krizhevsky, 2009) and ImageNet $64 \times 64$ (Deng et al., 2009). We measure the sample quality using Fréchet Inception Distance (FID) (Heusel et al., 2017) and Fréchet Distance under the DINOv2 model (Oquab et al., 2023) ($\text{FD}_{\text{DINOv2}}$) (Stein et al., 2024) and sampling efficiency using the number of function evaluations (NFEs). We also indicate the relative training costs of each of these methods. Implementation details can be found in Appendix D.

### 4.1 COMPARISON OF TRAINING SCHEMES

We compare CMs trained with ECT (denoted as ECM) against state-of-the-art diffusion models, advanced samplers for diffusion models, distillation methods such as consistency distillation (CD) (Song et al., 2023), and improved Consistency Training (iCT) (Song and Dhariwal, 2023). We show the training FLOPs, inference cost, and generative performance of the four training schemes.

**Score SDE/Diffusion Models.** We compare ECMs against Score SDE (Song et al., 2021b), EDM (Karras et al., 2022), and EDM with DPM-Solver-v3 (Zheng et al., 2024). 2-step ECM, which has been fine-tuned for only 100k iterations, matches Score SDE-deep, with 2× model depth and 2000 NFEs, in terms of FID. As noted in Fig. 2, ECM only requires a fraction of its inference cost and latency to achieve the same sample quality. 2-step ECM fine-tuned for 100k iterations outperforms EDM (Karras et al., 2022) with advanced DPM-Solver-v3 (NFE=10).

**Diffusion Distillation.** We compare ECT against Consistency Distillation (CD) (Song et al., 2023), a SoTA approach that distills a pretrained DM into a CM. As shown in Tab. 1, ECM significantly outperforms CD on both CIFAR-10 and ImageNet 64×64. We note that ECT is free from the errors of teacher DM and does not incur any additional cost of running teacher DM. 2-step ECM outperforms 2-step CD (with LPIPS (Zhang et al., 2018)) in terms of FID (2.20 vs 2.93) on CIFAR-10 while using around 1/3 of training compute of CD.

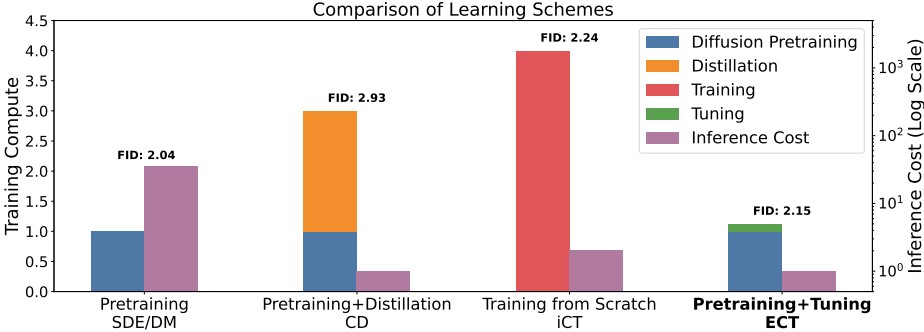

Figure 2: Comparison of training schemes for the diffusion-consistency family on CIFAR-10. Without relying on distillation from frozen diffusion teachers or extra adversarial supervision, ECT surpasses Consistency Distillation (CD) (Song et al., 2023) and Consistency Models trained from scratch (iCT) (Song and Dhariwal, 2023) using $1/4$ of the total training cost. ECT significantly reduces the inference cost compared to Score SDE/DMs (Diffusion Pretraining) while maintaining comparable sample quality.

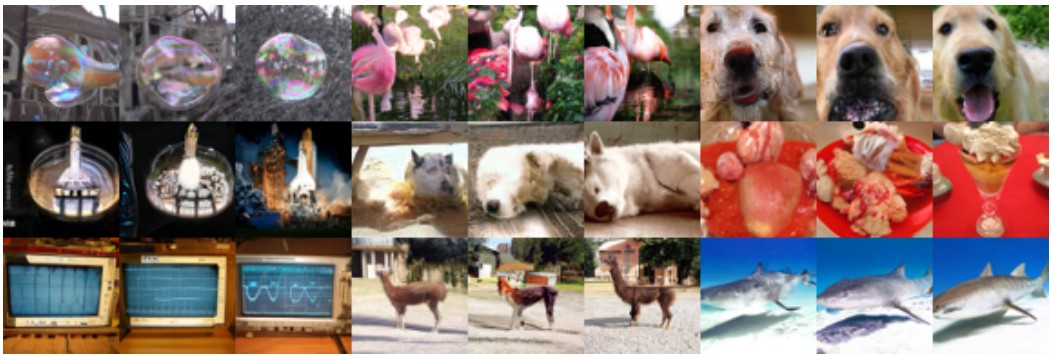

Figure 3: Scaling up training compute and model sizes results in improved sample quality on ImageNet $64 \times 64$. Each triplet (left-to-right) shows 2-step samples from ECM-S trained with $12.8$M images, ECM-S trained with $102.4$M images, and ECM-XL trained with $102.4$M images.

**Consistency training from scratch.** Improved Consistency Training (iCT) (Song and Dhariwal, 2023) is the SoTA recipe for training a consistency model from scratch without inferring the diffusion teacher. Compared to training from scratch, ECT rivals iCT-deep using $1/4$ of the overall training compute ($1/8$ in the tuning stage) and $1/2$ of the model size as shown in Fig. 2 and Tab. 1.

## 4.2 SCALING LAWS OF ECT

We leverage the efficiency of ECT to examine the scaling behavior of CMs, including training compute, model size, and model FLOPs. We find that when computational resources are not a bottleneck, ECT scales well and follows the classic power law.

**Training Compute.** Initializing from the weights of EDM (Karras et al., 2022), we fine-tune ECMs across six compute scales on CIFAR-10 (Krizhevsky, 2009) and plot the trend of FD$_{\text{DINOv2}}$ against training compute in Fig. 4 (Left). The largest compute reaches $2 \times$ the diffusion pretraining budget. As we scale up the training budget, we observe a classic power-law decay in FD$_{\text{DINOv2}}$, indicating that increased computational investment in ECT leads to substantial improvements in generative performance. Intriguingly, the gap between 1-step and 2-step generation becomes *narrower* when scaling up training compute, even while using the same $\Delta t \to 0$ schedule. We further fit the power-law FD$_{\text{DINOv2}} = K \cdot C^\alpha$, where $C$ is the normalized training FLOPs. The Pearson correlation coefficient between $\log(\text{Training Compute})$ and $\log(\text{FD}_{\text{DINOv2}})$ for 1-step and 2-step generation is $-0.9940$ and $-0.9996$, respectively, both with statistical significance ($p$-values $< 10^{-4}$).

Table 1: Generative performance on unconditional CIFAR-10 and class-conditional ImageNet 64×64. We use a budget of 12.8M training images (batch size 128 and 100k iterations) for ECMs. * stands for a budget of 102.4M training images (batch size 1024 and 100k iterations) on ImageNet 64×64. Results for prior methods are reported from Song and Dhariwal (2023); Karras et al. (2024).

**CIFAR-10**

| Method | FID↓ | NFE↓ |
|---|---|---|
| *Diffusion Models* | | |
| Score SDE (Song et al., 2020) | 2.38 | 2000 |
| Score SDE-deep (Song et al., 2020) | 2.20 | 2000 |
| EDM (Karras et al., 2022) | 2.01 | 35 |
| EDM (DPM-Solver-v3) (Zheng et al., 2024) | 2.51 | 10 |
| *Diffusion Distillation* | | |
| PD (Salimans and Ho, 2022) | 8.34 | 1 |
| GET (Geng et al., 2024) | 5.49 | 1 |
| Diff-Instruct (Luo et al., 2024) | 4.53 | 1 |
| TRACT (Berthelot et al., 2023) | 3.32 | 2 |
| CD (LPIPS) (Song et al., 2023) | 3.55 | 1 |
| CD (LPIPS) (Song et al., 2023) | 2.93 | 2 |
| *Consistency Models* | | |
| iCT (Song and Dhariwal, 2023) | 2.83 | 1 |
| | 2.46 | 2 |
| iCT-deep (Song and Dhariwal, 2023) | 2.51 | 1 |
| | 2.24 | 2 |
| *ECT* | | |
| ECM (100k iters) | 4.54 | 1 |
| ECM (200k iters) | 3.86 | 1 |
| ECM (400k iters) | 3.60 | 1 |
| ECM (100k iters) | 2.20 | 2 |
| ECM (200k iters) | 2.15 | 2 |
| ECM (400k iters) | 2.11 | 2 |

**ImageNet 64×64**

| Method | FID↓ | NFE↓ |
|---|---|---|
| *Diffusion Models* | | |
| ADM (Dhariwal and Nichol, 2021) | 2.07 | 250 |
| EDM (Karras et al., 2022) | 2.22 | 79 |
| EDM2-XL (Karras et al., 2023) | 1.33 | 63 |
| *Diffusion Distillation* | | |
| BOOT (Gu et al., 2023) | 16.3 | 1 |
| DFNO (LPIPS) (Zheng et al., 2023) | 7.83 | 1 |
| Diff-Instruct (Luo et al., 2024) | 5.57 | 1 |
| TRACT (Berthelot et al., 2023) | 4.97 | 2 |
| PD (LPIPS) (Salimans and Ho, 2022) | 5.74 | 2 |
| CD (LPIPS) (Song et al., 2023) | 4.70 | 2 |
| *Consistency Models* | | |
| iCT (Song and Dhariwal, 2023) | 3.20 | 2 |
| iCT-deep (Song and Dhariwal, 2023) | 2.77 | 2 |
| *ECT* | | |
| ECM-S (100k iters) | 3.18 | 2 |
| ECM-M (100k iters) | 2.35 | 2 |
| ECM-L (100k iters) | 2.14 | 2 |
| ECM-XL (100k iters) | 1.96 | 2 |
| ECM-S* | 4.05 | 1 |
| ECM-S* | 2.79 | 2 |
| ECM-XL* | 2.49 | 1 |
| ECM-XL* | 1.67 | 2 |

Table 2: Ablation study of design choices on ImageNet 64×64

| Methods | 1-step FID | 2-step FID |
|---|---|---|
| iCT + EDM2 Pretraining | 21.09 | 4.39 |
| + Continuous time schedule | 14.34 | 4.33 |
| + Dropout = 0.40 | 9.28 | 3.22 |
| + $\bar{w}(t) = 1/t^2 + 1/\sigma_{\text{data}}^2$ | 5.51 | 3.18 |

**Model Size & FLOPs.** Initialized from EDM2 (Karras et al., 2024) pretraining, we train ECM-S/M/L/XL models with parameters from 280M to 1.1B and model FLOPs from 102G to 406G. As demonstrated in Fig. 4, both 1-step and 2-step generation capabilities exhibit log-linear scaling for model FLOPs and parameters. This scaling behavior confirms that ECT effectively leverages increased model sizes and computational power to improve 1-step and 2-step generative capabilities.

Notably, ECT achieves *better* 2-step generation performance than state-of-the-art CMs, while utilizing only 33% of the overall computational budget compared to iCT (Song and Dhariwal, 2023) (batch size $4096 \times 800k$). This significant efficiency is achieved through a two-stage process: pretraining and tuning. While the pretraining stage utilizes the EDM2 pipeline, the tuning stage of ECT requires a remarkably modest budget of 12.8M training images (batch size $128 \times 100k$), ranging from 0.60% to 1.91% of the pretraining budget, depending on the model sizes.

**Inference.** Our scaling study also indicates a sweet spot for the inference of CMs. On both CIFAR-10 and ImageNet 64×64, there are 2-step inferences of smaller models surpassing 1-step inferences of larger models, *e.g.*, 498M ECM-M against 1.1B ECM-XL. This calls for further studies of inference-optimal scaling and test-time compute scaling for visual generation.

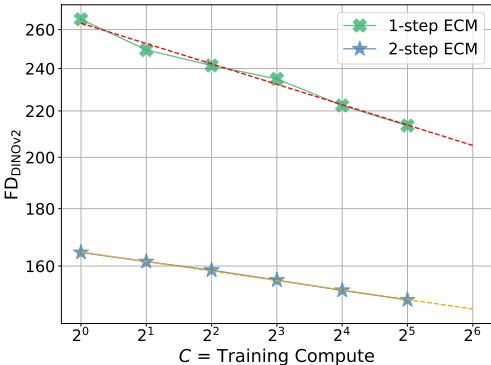 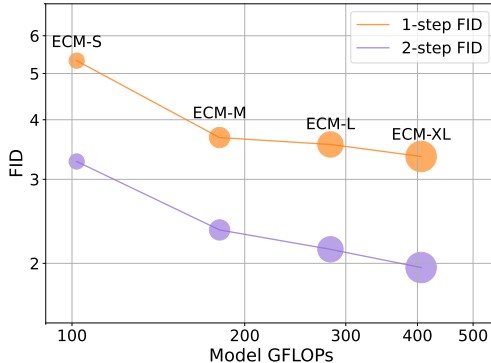

Figure 4: (Left): Scaling up training compute yields the classic power-law between $\mathrm{FD_{DINOv2}} \downarrow$ and training compute, with $K = 263$, $\alpha = -0.060$ for 1-step inference, and $K = 164$, $\alpha = -0.028$ for 2-step inference, (Right): Given the same batch size and iterations, scaling up model sizes and model FLOPs strongly correlates with FID $\downarrow$ improvements on ImageNet $64 \times 64$. The diameter is proportional to the model size.

### 4.3 SCALING TO HIGH-RESOLUTION DATASETS

We further extend our method to latent space and conducted experiments on ImageNet $512 \times 512$. Our ECM-M (498M parameters) achieves FID of 6.88 for 2-step generation using a modest budget of 12.8M training images, surpassing ADM-G (Dhariwal and Nichol, 2021)(559M parameters, FID of 7.72 at NFE=250x2) on the ImageNet $512 \times 512$ dataset. The strong performance on ImageNet 512×512 also indicates that the insights from our scaling analysis generalize beyond the low-resolution regime.

### 4.4 ABLATION STUDY

We perform ablation studies to analyze the impact of various tuning design choices. Using iCT's design choices as the baseline, we tune EDM2-S and present the results in Tab. 2. These results highlight the benefits of the continuous-time training schedule, increased dropout, and weighting functions in enhancing ECMs' efficient generative performance.

## 5 RELATED WORK

**Consistency Models.**   Consistency models (Song et al., 2023; Song and Dhariwal, 2023) are a new family of generative models designed for efficient generation with few model steps without the need for adversarial training. CMs do not rely on a pretrained diffusion model (DM) to generate training targets but instead leverage an unbiased score estimator. CMs have been extended to multi-step sampling (Kim et al., 2024; Wang et al., 2024b; Heek et al., 2024), latent space models (Luo et al., 2023), ControlNet (Xiao et al., 2023), video (Wang et al., 2024c), and combined with additional adversarial losses (Kim et al., 2024; Kong et al., 2023). Despite their sampling efficiency, CMs are typically more challenging to train and require significantly more compute resources compared to their diffusion counterparts. Our work substantially improves the training efficiency of CMs, reducing the cost of future research and deployment on CMs.

While initializing CMs with pretrained diffusion models has also been considered in Song et al. (2023); Song and Dhariwal (2023), their training schedules for discrete-time models neither correspond to the standard diffusion pretraining phase initially nor demonstrate the experimental advantages of this two-stage approach. While Continuous-time CT considers initializing from pretrained diffusion models, it is schedule-free, jumping from the initialization to the continuous-time training. ECT, starting from the pretraining stage, progressively reduces the discretization error using the continuous-time schedule.

ECT is considered a training method rather than a distillation approach since distillation needs running inference of a frozen teacher model to generate training signals online or offline. ECT requires neither

of these. It utilizes a single network (and maintains its Exponential Moving Average (EMA) copy) throughout the pretraining and tuning stage. The division between pretraining and tuning under the continuous-time schedule is more conceptual than technical since they are approximations at different degrees of consistency models.

**Diffusion Distillation.** Drawing inspiration from knowledge distillation (Hinton et al., 2015), distillation is the most widespread training-based approach to accelerate the diffusion sampling procedure. In diffusion distillation, a pretrained diffusion model (DM), which requires hundreds to thousands of model evaluations to generate samples, acts as a teacher. A student model is trained to match the teacher model's sample quality, enabling it to generate high-quality samples in a few steps.

There are two main lines of work in this area. The first category involves trajectory matching, where the student learns to match points on the teacher's sampling trajectory. Methods in this category include offline distillation (Luhman and Luhman, 2021; Geng et al., 2024; Zheng et al., 2023), which require an offline synthetic dataset generated by sampling from a pretrained DM to distill a teacher model into a few-step student model; progressive distillation (Salimans and Ho, 2022; Meng et al., 2023), and TRACT (Berthelot et al., 2023), which require multiple training passes or offline datasets to achieve the same goal; and BOOT (Gu et al., 2023), Consistency Distillation (CD)(Song et al., 2023), and Imagine-Flash (Kohler et al., 2024), which minimize the difference between the student predictions at carefully selected points on the sampling trajectory.

CD is closely related to our method, as it leverages a teacher model to generate pairs of adjacent points and enforces the student predictions at these points to map to the initial data point. However, it employs a fixed schedule derived from a specific sampler, which may introduce non-negligible discretization errors in approximating the consistency condition. It also limits the quality of consistency models to that of the pretrained diffusion model.

The second category minimizes the probabilistic divergence between data and model distributions, *i.e.,* distribution matching (Poole et al., 2022; Wang et al., 2024a; Luo et al., 2024; Yin et al., 2023; Zhou et al., 2024a). These methods (Luo et al., 2024; Sauer et al., 2023; Yin et al., 2023; Nguyen and Tran, 2023; Kohler et al., 2024; Xu et al., 2023a; Lin et al., 2024; Zhou et al., 2024b) use score distillation or adversarial loss, to distill an expensive teacher model into an efficient student model. However, they can be challenging to train in a stable manner due to the alternating updating schemes from either adversarial or score distillation. Some of these methods such as DreamFusion (Poole et al., 2022) and ProlificDreamer (Wang et al., 2024a) are used for 3D object generation.

A drawback of training-based approaches is that they need additional training procedures after pretraining to distill an efficient student, which can be computationally intensive. For a detailed discussion on the recent progress of diffusion distillation, we direct the readers to Dieleman (2024).

**Fast Samplers for Diffusion Models.** Fast samplers are usually training-free and use advanced solvers to simulate the diffusion stochastic differential equation (SDE) or ordinary differential equation (ODE) to reduce the number of sampling steps. These methods reduce the discretization error during sampling by analytically solving a part of SDE or ODE (Lu et al., 2022a; Xue et al., 2024; Lu et al., 2022b), by using exponential integrators and higher order polynomials for better approximation of the solution (Zhang and Chen, 2022), using higher order numerical methods (Karras et al., 2022), using better approximation of noise levels during sampling (Kong and Ping, 2021), correcting predictions at each step of sampling (Zhao et al., 2024) and ensuring that the solution of the ODE lies on a desired manifold (Liu et al., 2022a). Another orthogonal strategy is the parallel sampling process (Pokle et al., 2022; Shih et al., 2024), solving fixed points of the entire trajectory. A drawback of these fast samplers is that the quality of samples drastically reduces as the number of sampling steps goes below a threshold such as 10 steps.

## 6 CONCLUSION

We present Easy Consistency Tuning (ECT), a simple yet efficient scheme for training consistency models. The resulting models, ECMs, unlock state-of-the-art few-step generative capabilities at a minimal tuning cost and are able to benefit from scaling. Our code is available for future prototyping, studying, and deploying consistency models within the community.

## BROADER IMPACTS AND ETHICS STATEMENT

We propose Easy Consistency Tuning (ECT) that can efficiently train consistency models as state-of-the-art few-step generators, using only a small fraction of the computational requirements compared to current CMs training and diffusion distillation methods. We hope that ECT will democratize the creation of high-quality generative models, enabling artists and creators to produce content more efficiently. While this advancement can aid creative industries by reducing computational costs and speeding up workflows, it also raises concerns about the potential misuse of generative models to produce misleading, fake, or biased content. We conduct experiments on academic benchmarks, whose resulting models are less likely to be misused. Further experiments are needed to better understand these consistency model limitations and propose solutions to address them.

## REPRODUCIBILITY STATEMENT

We provide extensive details of experimental settings and hyperparameters to reproduce our experimental results in Appendix D. We have released our code to ensure transparency and reproducibility of the results.

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

## A  MOTIVATIONS BEHIND DESIGN CHOICES IN ECT

In this section, we expand upon our motivation behind the design decisions for the mapping function, metric, and weighting function used for ECT.

**Mapping Function.**   We first assume that $\Delta t$ is *approximately* proportional to $t$. Let $0 < c \le 1$ be this constant of proportionality, then we can write:

$$c \approx \frac{\Delta t}{t} = \frac{t - r}{t} = 1 - \frac{r}{t} \Rightarrow \frac{r}{t} \approx 1 - c.$$

As training progresses, the mapping function should gradually shrink $\Delta t \to 0$. However, the above parameterization does not achieve this. An alternative parameterization is to decrease $\Delta t$ exponentially. We assume the ratio between $r$ and $t$ can be written as:

$$\frac{r}{t} = 1 - \frac{1}{q^a}, \tag{17}$$

where $q > 1$, $a = \lfloor \text{iters}/d \rfloor$, and $d$ is a hyperparameter controlling how quickly $\Delta t \to \mathrm{d}t$. At the beginning of training, $\frac{r}{t} = 1 - \frac{1}{q^0} = 0 \Rightarrow r = 0$, which falls back to DMs. Since we can initialize from the diffusion pretraining, this stage can be skipped by setting $a = \lceil \text{iters}/d \rceil$. As training progresses (iters $\uparrow$), $\frac{r}{t} \to 1$ leads to $\Delta t \to \mathrm{d}t$.

Finally, we adjust the mapping function to balance the prediction difficulties across different noise levels:

$$\frac{r}{t} = 1 - \frac{1}{q^a} n(t) = 1 - \frac{1}{q^{\lceil \text{iters}/d \rceil}} n(t). \tag{18}$$

For $n(t)$, we choose $n(t) = 1 + k, \sigma(-b, t) = 1 + \frac{k}{1 + e^{bt}}$, using the sigmoid function $\sigma$. Since $r \ge 0$, we also clamp $r$ to satisfy this constraint after the adjustment.

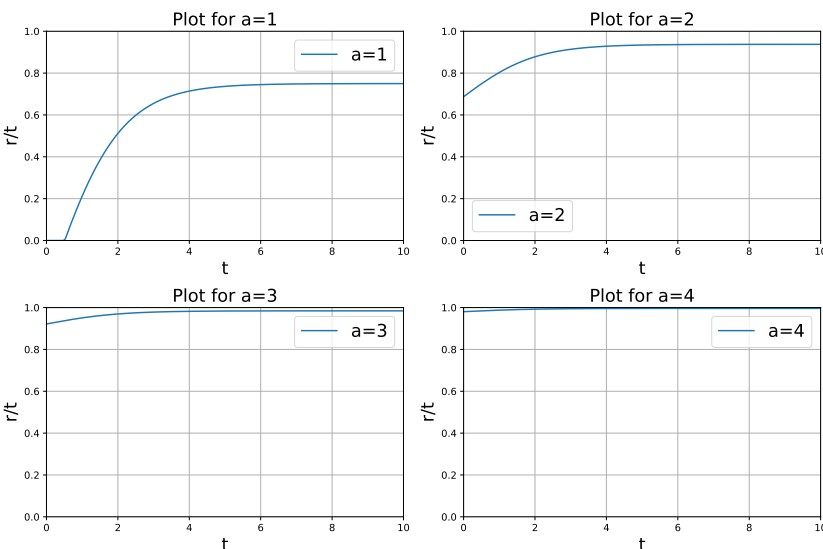

Figure 5: Visualization of $r/t$ during training. $\Delta t \to \mathrm{d}t$ when $r/t \to 1$.

The intuition behind this mapping function is that the relative difficulty of predicting $f(\mathbf{x}_r)$ from $\mathbf{x}_t$ can vary significantly across different noise levels $t$ when using a linear mapping between $t$ and $r$.

Consider $r/t = 0.9$. At small values of $t$, $\mathbf{x}_t$ and $\mathbf{x}_r$ are close, making the alignment of $f(\mathbf{x}_t)$ with $f(\mathbf{x}_r)$ relatively easy. In contrast, at larger $t$, where $\mathbf{x}_t$ and $\mathbf{x}_r$ are relatively far apart, the distance between the predictions $f(\mathbf{x}_t)$ and $f(\mathbf{x}_r)$ can be substantial. This leads to imbalanced gradient flows across different noise levels, impeding the training dynamics.

Therefore, we downscale $r/t$ when $t$ is near $0$ through the mapping function, balancing the gradient flow across varying noise levels. This prevents the gradient at any noise level from being too small or too large relative to other noise levels, thereby controlling the variance of the gradients.

We direct the reader to Appendix B for details of how to set $q^{\lceil \text{iters}/d \rceil}$.

**Choice of Metric.** iCT uses pseudo-Huber metric (Charbonnier et al., 1997) to mitigate the perceptual bias caused by the LPIPS metric (Zhang et al., 2018),

$$L(\mathbf{x}, \mathbf{y}) = \sqrt{\|\mathbf{x} - \mathbf{y}\|_2^2 + \epsilon} - \epsilon, \quad \epsilon > 0. \tag{19}$$

This metric indeed improves the performance of CMs over the classic squared $L_2$ loss. When taking a careful look as this metric, we reveal that one of the reasons for this improvement is that this metric is more robust to the outliers compared to the $L_2$ metric due to its adaptive per-sample scaling of the gradients. Let $\Delta = \mathbf{x} - \mathbf{y}$, then the differential of the pseudo-Huber metric can be written as

$$\mathrm{d}L = \underbrace{\frac{1}{\sqrt{\|\Delta\|_2^2 + c^2}}}_{\text{weighting term}} \underbrace{\mathrm{d}\left(\frac{1}{2}\|\Delta\|_2^2\right)}_{\text{differential of squared } L_2 \text{ loss}}, \tag{20}$$

where we have decomposed the differential of pseudo-Huber loss into an adaptive weighting term and the differential of the squared $L_2$ loss. Therefore, we retain the squared $L_2$ metric used in DMs, and explore varying adaptive weighting terms which we explore in detail in Appendix B.

**Distinction between the training schedules of ECT and iCT.** As noted in Sec. 2, iCT (Song and Dhariwal, 2023) employs a discrete-time curriculum given by Eq. (10). This curriculum divides the noise horizon $[0, T]$ into $N$ smaller consecutive subintervals to apply the consistency loss, characterized by non-overlapping segments $[t_i, t_{i+1}]$, and gradually increases the number of intervals $N = 10 \rightarrow 1280$. However, the "boundary" condition of this schedule is to start with the number of intervals to $N = 1$, learning a model solely mapping samples at noise levels $T_{\max}$ to the clean data $\mathbf{x}_0$, largely distinct from the classic diffusion models training. We instead investigate a continuous-time schedule whose "boundary" condition yields diffusion pretraining, *i.e.,* constructing training pairs of $r = 0$ for all $t$ at the beginning.

# B   EXPLORING DESIGN SPACE & SCALING OF CONSISTENCY MODELS

Due to ECT's efficiency, we can explore the design space of CMs at a minimal cost. We specifically examine the weighting function, training schedule, and regularization for CMs.

Our most significant finding is that *controlling gradient variances and balancing the gradients across different noise levels are fundamental to CMs' training dynamics*. Leveraging the deep connection between CMs and DMs, we also improve the diffusion pretraining and the full pretraining+tuning pipeline using our findings.

**Weighting Function.** Forward processes with different noise schedules and model parameterizations can be translated into each other at the cost of varying weighting functions (Kingma and Gao, 2024). From our experiments on a wide range of weighting schemes, we learn three key lessons.

(1) *There is no free lunch for weighting function*, *i.e.,* there is likely no universal timestep weighting $\bar{w}(t)$ that can outperform all other candidates on different datasets, models, and target metrics for both 1-step and 2-step generation.

We refer these results to Tab. 3, including $\mathrm{SNR}(t) = 1/t^2$, $\mathrm{SNR}(t) + 1 = 1/t^2 + 1$ (Salimans and Ho, 2022), EDM weighting $\mathrm{SNR}(t) + 1/\sigma_{\text{data}}^2 = 1/t^2 + 1/\sigma_{\text{data}}^2$ (Karras et al., 2022), and Soft-Min-SNR weighting $1/(t^2 + \sigma_{\text{data}}^2)$ (Hang et al., 2023; Crowson et al., 2024), where $\mathrm{SNR}(t) = 1/t^2$ is the signal-to-noise ratio in our setup.

On CIFAR-10, the weighting $\bar{w}(t) = 1/(t-r)$ from the discretization of consistency condition in Eq. (12) achieves the best 1-step FID, while the square root of $\mathrm{SNR}(t)$, $\bar{w}(t) = \sqrt{\mathrm{SNR}(t)} = 1/t$, produces the best $\mathrm{FD}_{\text{DINOv2}}$. On ImageNet $64 \times 64$, considering that we have already had the adaptive

Table 3: Performance of ECMs trained with various weighting functions on ImageNet 64×64. We enable the adaptive weighting $w(\Delta) = 1/(\|\Delta\|_2^2 + c^2)^{\frac{1}{2}}$.

| $\bar{w}(t)$ | 1-step FID↓ | 2-step FID↓ |
|---|---|---|
| $1$ | **5.39** | 3.48 |
| $1/t$ | 17.79 | 3.24 |
| $1/(t-r)$ | 9.28 | 3.22 |
| $1/t + 1/\sigma_{\text{data}}$ | 5.68 | 3.44 |
| $1/t^2$ | 190.80 | 20.65 |
| $1/t^2 + 1$ | 6.78 | **3.12** |
| $1/t^2 + 1/\sigma_{\text{data}}^2$ | 5.51 | 3.18 |
| $1/(t^2 + \sigma_{\text{data}}^2)$ | 163.01 | 13.33 |

Table 4: Performance of ECMs trained with varying adaptive weightings on ImageNet 64×64.

| $\bar{w}(t)$ | $w(\Delta)$ | 1-step FID↓ | 2-step FID↓ |
|---|---|---|---|
| $1/t^2 + 1/\sigma_{\text{data}}^2$ | $1$ | 6.51 | 3.28 |
| $1/t^2 + 1/\sigma_{\text{data}}^2$ | $1/(\|\Delta\|_1 + c)$ | 6.29 | 3.25 |
| $1/t^2 + 1/\sigma_{\text{data}}^2$ | $1/(\|\Delta\|_2^2 + c^2)^{\frac{1}{2}}$ | 5.51 | 3.18 |
| $1$ | $1/(\|\Delta\|_2^2 + c^2)^{\frac{1}{2}}$ | 5.39 | 3.48 |

weighting $w(\Delta)$, the uniform weighting $\bar{w}(t) \equiv 1$ can demonstrate the best 1-step FID when tuning from EDM2 (Karras et al., 2024). In contrast to 1-step FIDs, a wider range of timestep weighting $\bar{w}(t)$ produces close 2-step FIDs for ECMs.

When starting on a new dataset with no prior information, $\bar{w}(t) = \text{SNR}(t) + n$ is a generally strong choice as the default timestep weighting of data prediction models (x-pred). In this situation, this weighting function corresponds to using v-pred (Salimans and Ho, 2022) or flow matching (Lipman et al., 2022; Liu et al., 2022b) as model parameterization when $n = 1$.

(2) *The adaptive weighting $w(\Delta)$ achieves better results by controlling gradient variance.* The adaptive weighting $w(\Delta)$ on a per-sample basis shows uniform improvements on both CIFAR-10 and ImageNet 64×64. See Tab. 4 for the ablation study.

Beyond ECT, we further investigate the role of adaptive weighting $w(\Delta)$ in pretraining on a toy Swiss roll dataset using the parameterization and forward process of flow matching (Lipman et al., 2022) and an MLP network.

Consider the objective function $w(\Delta)\|v_\theta(\mathbf{x}_t) - (\mathbf{x}_1 - \mathbf{x}_0)\|_2^2$, where $\mathbf{x}_t = (1 - t) \cdot \mathbf{x}_0 + t \cdot \mathbf{x}_1$, $t \sim \text{Uniform}(0, 1)$, $\mathbf{x}_1 \sim \mathcal{N}(\mathbf{0}, \mathbf{I})$, and the adaptive weighting

$$w(\Delta) = \frac{1}{(\|\Delta\|_2^2 + c^2)^p},$$

where $p = 0$ corresponds to no adaptive weighting. We set $c^2 = 10^{-6}$ and control the strength of gradient normalization by varying $p$ from 0 to 1.

As we increase the strength of adaptive weighting, flow models become easier to sample from in a few steps. Surprisingly, even $p = 1$ demonstrates strong few-step sampling results when pretraining the flow model. See Fig. 6 for visualization.

**Mapping Function.** We compare the constant mapping function with $n(t) \equiv 1$ in Eq. (17) and mapping function equipped with the sigmoid $n(t)$ in Eq. (18). We use $k = 8$ and $b = 1$ for all the experiments, which transfers well from CIFAR-10 to ImageNet 64×64 and serves as a baseline in our experiments. Though $b = 2$ can further improve the 1-step FIDs on ImageNet 64×64, noticed post hoc, we don't rerun our experiments.

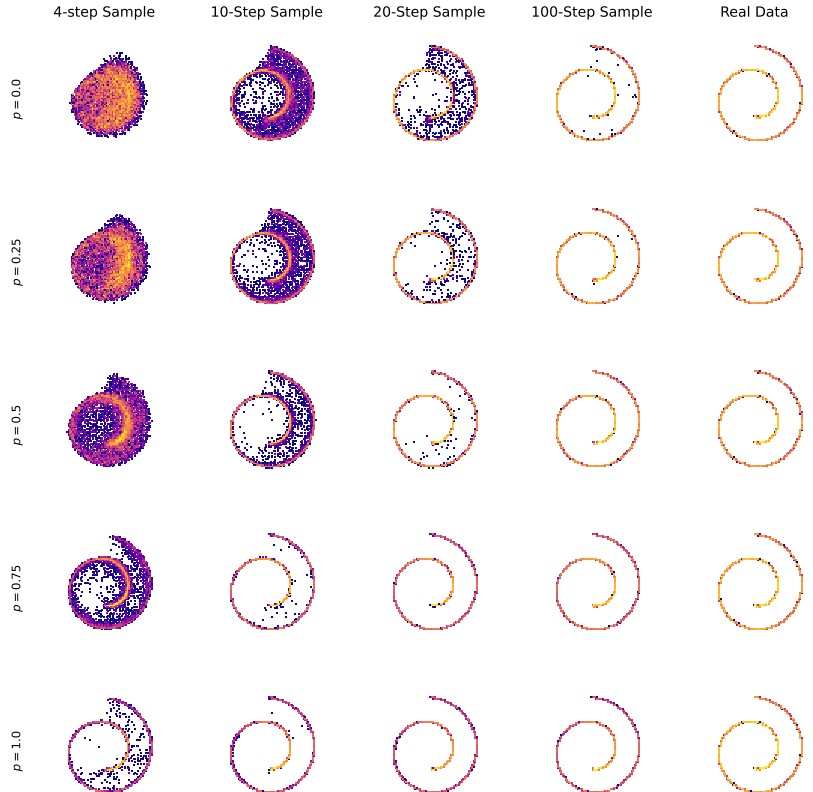

Figure 6: Influence of adaptive weighting $w(\Delta) = 1/(\|\Delta\|_2^2 + \epsilon)^p$ on pretraining using varying $p$.

On CIFAR-10, the constant mapping function with $n(t) \equiv 1$ achieves 1-step FID of 4.06 at 200k iterations, worse than the 1-step FID of 3.86 by $n(t) = 1 + \frac{k}{1+e^{bt}}$. Under our forward process $(\mathbf{x}_t = \mathbf{x}_0 + t \cdot \epsilon)$ and model parameterization (EDM (Karras et al., 2022)), the constant mapping function incurs training instability on ImageNet 64×64, likely due to the imbalanced gradient flow.

The role of the mapping function, regarding training, is to balance the difficulty of learning *consistency condition* across different noise levels, avoiding trivial consistency loss near $t \to 0$. For model parameterizations and forward processes different from ours, for example, flow matching (Lipman et al., 2022; Liu et al., 2022b; Sauer et al., 2024), we advise readers to start from the constant mapping function due to its simplicity.

**Dropout.** In line with (Song and Dhariwal, 2023), we find that CMs benefit significantly from dropout (Hinton et al., 2012). On CIFAR-10, we apply a dropout of 0.20 for models evaluated on FID and a dropout of 0.30 for models evaluated on $FD_{DINOv2}$.

On ImageNet 64×64, we note that ECT benefits from a surprisingly high dropout rate. When increasing the dropout rate from 0.10 to 0.40, the 2-step FID decreases from 4.53 to 3.24. Increasing the dropout rate further can be helpful for 1-step FID under certain timestep weighting $\bar{w}(t)$, but the 2-step FID starts to deteriorate. In general, we optimize our model configurations for 2-step generation and choose the dropout rate of 0.40 for ECM-S.

Finally, we note that the dropout rate tuned at a given weighting function $w(t)$ transfers well to the other weighting functions, thereby reducing the overall cost of hyperparameter tuning. On ImageNet 64×64, the dropout rate can even transfer to different model sizes. We apply a dropout rate of 0.50 for all the model sizes of ECM-M/L/XL.

Table 5: Generative performance on class-conditional CIFAR-10.

| Method | FD$_{\text{DINOv2}}\downarrow$ | NFE$\downarrow$ |
|---|---|---|
| GANs | | |
| BigGAN (Brock et al., 2018) | 326.66 | 1 |
| StyleGAN2-ADA (Karras et al., 2020) | 305.92 | 1 |
| StyleGAN-XL (Sauer et al., 2022) | 204.60 | 1 |
| Diffusion Models | | |
| EDM (Karras et al., 2022) | 145.20 | 35 |
| PFGM++ (Xu et al., 2023b) | 141.65 | 35 |
| ECT | | |
| ECM (ECT Pretrained) | 121.05 | 35 |
| ECM (Tuned) | 198.51 | 1 |
| ECM (Tuned) | 128.63 | 2 |

**Shrinking** $\Delta t \to 0$. In the mapping function discussed in Sec. 3.2 and Appendix A, we use the hyperparameter $d$ to control the magnitude of $q$, thereby determining the overall rate of shrinking $\Delta t \to 0$, given by $\left(1 - 1/q^{\lceil \text{iters}/d \rceil}\right)$. In practice, we set $q = 2$ and $d = \text{total\_iters}//8$ for CIFAR-10 experiments, and $q = 4$ and $d = \text{total\_iters}//4$ for ImageNet 64×64 experiments, achieving $r/t \approx 0.99$ at the end of training.

Compared with no shrinkage of $\Delta t$, where $\Delta t \approx \mathrm{d}t$ throughout, we find that shrinking $\Delta t \to 0$ results in improved performance for ECMs. For example, on CIFAR-10, starting ECT directly with $\Delta t \approx \mathrm{d}t$ by setting $q = 256$ (corresponding to $r/t \approx 0.99$) leads to quick improvements in sample quality initially but slower convergence later on. The 1-step FID drops from 3.60 to 3.86 using the same 400k training iterations compared to gradually shrinking $\Delta t \to 0$. On ImageNet 64×64, $\Delta t \approx \mathrm{d}t$ with $q = 256$ from the beginning results in training divergence, as the gradient flow is highly imbalanced across noise levels, even when initializing from pretrained diffusion models.

This observation suggests that *ECT's schedule should be adjusted according to the compute budget*. At small compute budgets, as long as training stability permits, directly approximating the *differential consistency condition* through a small $\Delta t \approx \mathrm{d}t$ leads to fast sample quality improvements. For normal to rich compute budgets, shrinking $\Delta t \to 0$ generally improves the final sample quality, which is the recommended practice.

Using this feature of ECT, we demonstrate its efficiency by training ECMs to surpass previous Consistency Distillation, which took hundreds of GPU hours, using **one hour on a single A100 GPU**.

**Training Generative Models in 1 GPU Hour.** Deep generative models are typically computationally expensive to train. Unlike training a classifier on CIFAR-10, which usually completes within one GPU hour, leading generative models on CIFAR-10 as of 2024 require days to a week to train on 8 GPUs. Even distillation from pretrained diffusion models can take over a day on 8 GPUs or even more, equivalently hundreds of GPU hours.

To demonstrate the efficiency of ECT and facilitate future studies, we implemented a fast prototyping setting designed to yield results within one hour on a single GPU. This configuration uses a fixed $\Delta t \approx \mathrm{d}t$ by setting $q = 256$ in our mapping function (corresponding to $\frac{r}{t} \approx 0.99$), which allows for quick approximation of the differential consistency condition. Through 8000 gradient descent steps at batch size of 128, within 1 hour on a single A100 40GB GPU, ECT achieves a 2-step FID of 2.73, outperforming Consistency Distillation (2-step FID of 2.93) trained with 800k iters at batch size 512 and LPIPS (Zhang et al., 2018) metric.

**Importance of Pretraining** We conducted a controlled experiment, training CMs from scratch following iCT best practices and tuning EDM using both iCT and ECT schedules. The combined pretraining and fine-tuning cost for ECT is $50\% + 6.25\%$ of iCT trained from scratch. Results are presented in Tab. 6.

This performance gap is particularly noticeable when evaluating models using modern metrics like $\text{FD}_{\text{DINOv2}}$ (Stein et al., 2024). $\text{FD}_{\text{DINOv2}}$ uses the representation space of the DINOv2-L model to compute distributional distance, which has been shown to better align with human evaluations.

Table 6: Impact of pretraining on model performance

| Model | $\text{FD}_{\text{DINOv2}}$ |
|---|---|
| iCT from scratch | 242.30 |
| Pretraining + iCT tuning | 200.31 |
| Pretraining + ECT tuning | 190.13 |
| EDM | 168.16 |

When scaling up the pretraining+tuning cost to match the overall cost of iCT in class-conditional settings, ECM achieves a $\text{FD}_{\text{DINOv2}}$ of 152.21, significantly outperforming the iCT model trained from scratch (205.11). For context, StyleGAN-XL achieves an $\text{FD}_{\text{DINOv2}}$ of 204.60.

**Improving Pretraining using Findings in Tuning.** The exploration of the design space through the tuning stage as a proxy led to a question: Can the insights gained during tuning be applied to improve the pretraining stage and, consequently, the entire pretraining+tuning pipeline for CMs? The results of our experiments confirmed this hypothesis.

For the largest $\Delta t = t$, ECT falls back to diffusion pretraining with $r = 0$ and thus $f_\theta(\mathbf{x}_r) = \mathbf{x}_0$. We pretrain EDM (Karras et al., 2022) on the CIFAR-10 dataset using the findings in ECT. Instead of using EDM weighting, $\text{SNR}(t) + 1/\sigma_{\text{data}}^2$, we enable the adaptive weighting $w(\Delta)$ with $p = 1/2$ and smoothing factor $c = 0$ and a timestep weighting $\bar{w}(t) = 1/t$.

Compared with the EDM baseline, the recipe from ECT brings a convergence acceleration over $2\times$ regarding $\text{FD}_{\text{DINOv2}}$, matching EDM's final performance using less than half of the pretraining budget and largely outperforming it at the full pretraining budget.

EDM pretrained by ECT achieves $\text{FD}_{\text{DINOv2}}$ of 150.39 for unconditional generation and 121.05 for class-conditional generation, considerably better than the EDM baseline's $\text{FD}_{\text{DINOv2}}$ of 168.17 for unconditional generation and 145.20 for class-conditional generation, when using the same pretraining budget and inference steps (NFE=35).

**Influence of Pretraining Quality.** Using ECT pretrained models ($\text{FD}_{\text{DINOv2}}$ of 121.05) and original EDM (Karras et al., 2022) ($\text{FD}_{\text{DINOv2}}$ of 145.20), we investigate the influence of pretraining quality on consistency tuning and resulting ECMs. Our experiments confirm that better pretraining leads to easier consistency tuning and faster convergence. In the same budget of 204.8M images, tuning from ECT pretrained models achieves $\text{FD}_{\text{DINOv2}}$ of 128.63, better than $\text{FD}_{\text{DINOv2}}$ of 152.21 from EDM.

ECM from the ECT pretraining surpasses SoTA GANs in 1 sampling step and advanced DMs in 2 sampling steps, only slightly falling behind our pretrained models and setting up a new SoTA for the modern metric $\text{FD}_{\text{DINOv2}}$. See Tab. 5 for details.

On ImageNet 64×64, ECM-M, initialized from EDM2-M (Karras et al., 2024), deviates from the power law scaling and achieves better generative performance than the log-linear trend. (See Fig. 4, Right). We speculate that it is due to a higher pretraining budget, in which EDM2-M was pretrained by $2\times$ training images compared with other model sizes (S/L/XL).

**Differences between 1-step and 2-step Generation.** Our empirical results suggest that the training recipe for the best 1-step generative models can differ from the best few-step generative models in many aspects, including weighting function, dropout rate, and EMA rate/length. Fig. 7 (Left) shows an example of how FIDs from different numbers of function evaluations (NFEs) at inference vary with dropout rates.

In our setups, starting from a proper model size, the improvements from 2-step sampling seem larger than doubling the model size but keeping 1-step sampling. In the prior works, iCT (Song and Dhariwal, 2023) employs $2\times$ deeper model, but the 1-step generative performance can be inferior to the 2-step results from ECT. This finding is consistent with recent theoretical analysis (Lyu et al.,

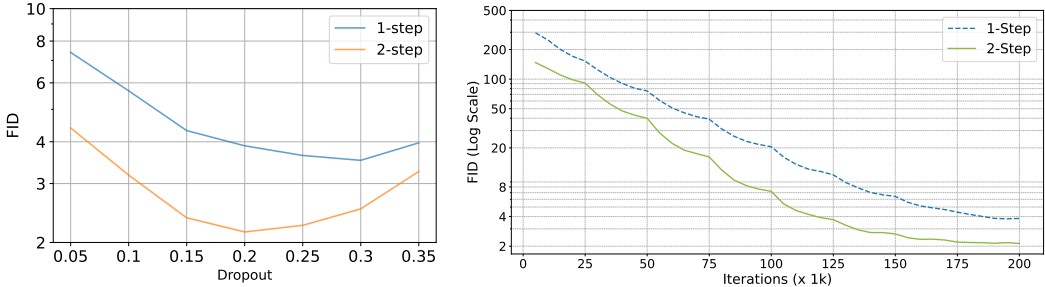

Figure 7: (Left): Relationship between the dropout and FIDs for models trained on CIFAR-10 with varying numbers of function evaluations (NFE) at inference. (Right): Evolution of model performance w.r.t. training iterations.

2023), which indicates a tighter bound on the sample quality for the 2-step generation compared to the 1-step generation.

**Pareto Frontier & Scaling Law.** The Pareto Frontier reveals a seemingly power law scaling behavior. Training configurations not optimized for the current compute budget, *i.e.,* not on the Pareto Frontier, deviate from this scaling. Simply scaling up the training compute without adjusting other parameters may result in suboptimal performance. In our compute scaling experiments, we increased the batch size and enabled the smoothing factor $c$ in the adaptive weighting to maintain this trend.

**Performance Evolution along Training.** We visualize the performance evaluation along the training process in Fig. 7 (Right). When the approximation errors of the consistency condition are reduced, the sample quality improves correspondingly.

## C   EXTENSION TO CONSISTENCY DISTILLATION.

**Easy Consistency Distillation** Consistency Distillation (CD) (Song et al., 2023) uses a fixed discrete schedule derived from a specific sampler, such as the EDM sampler. This approach inherently limits $\Delta t \to \mathrm{d}t$, therefore causing a non-trivial discretization error on approximating the differential consistency condition. Building upon ECT, we extend our continuous-time schedule to Consistency Distillation, which we term Easy Consistency Distillation (ECD).

On ImageNet 64×64, we implement Consistency Distillation as the baseline using pretrained EDM2-S (Karras et al., 2024) with weighting functions, noise distribution, and dropout. The results, shown in Tab. 7, demonstrate that continuous-time distillation improves upon the standard CD approach.

Given pretrained DMs, ECD typically demonstrates a performance advantage over ECT at smaller batch sizes (e.g., 128), primarily due to the variance reduction in approximating the score function:

$$\nabla_{\mathbf{x}_t} \log p(\mathbf{x}_t) = \mathbb{E}[\nabla_{\mathbf{x}_t} \log p(\mathbf{x}_t|\mathbf{x}_0)|\mathbf{x}_t]$$

This advantage stems from the teacher models, which provide a reduced variance estimate compared to the Monte Carlo estimation used in ECT. This gap tends to narrow as we scale up to larger model sizes and training budget. Scaling compensates for the limited training budget and variances, closing the gap between the two.

Consequently, while ECD may outperform ECT in resource-constrained scenarios or with smaller models, the distinction becomes less pronounced as we move to larger scales. This observation underscores the scaling factor (computational resources and model size) when choosing between ECT and ECD for a given application.

**Data-Free ECD.** Inspired by the recent progress on data-free distillation (Luo et al., 2024; Gu et al., 2023; Yin et al., 2023; Zhou et al., 2024a), we explore a data-free variant of Easy Consistency Distillation (ECD). Rather than sampling data points $\mathbf{x}_0$ from a given dataset $\mathcal{D}$, we generate synthetic data points directly from the CM itself on the fly, $\mathbf{x}_0 = f_{\mathrm{sg}(\theta)}(\boldsymbol{\epsilon}', T)$, where $\boldsymbol{\epsilon}' \sim p(\boldsymbol{\epsilon})$, $T$ is the

Table 7: Easy Consistency Distillation (ECD) on ImageNet 64×64 using the same budget of 12.8M training images (batch size 128 and 100k iterations) as ECT in Tab. 1.

| Methods | 1-step FID | 2-step FID |
|---------|-----------|-----------|
| CD-S | 8.18 | 3.71 |
| ECD-S | 3.33 | 2.10 |
| ECD-M | 2.78 | 1.92 |
| ECD-XL | 2.54 | 1.77 |

maximum noise level for sampling, followed by the same ECD training loss over the self synthetic data. This eliminates the need for a distillation dataset or the construction of synthetic data from the teacher model (Luhman and Luhman, 2021; Geng et al., 2024; Zheng et al., 2023).

In ImageNet 64×64, this data-free ECD achieves a 1-step FID of 4.38 and a 2-step FID of 2.77, comparable to the data-dependent ECD and ECT schemes in the same budget. It suggests that data-free ECD can be a competitive alternative in scenarios where access to large datasets is limited or unavailable. We summarize both ECD and its data-free variant in Alg. 2.

---

**Algorithm 2** Easy Consistency Distillation (ECD)

**Input:** Dataset $\mathcal{D}$ (for ECD), a pretrained diffusion model $\phi$, mapping function $p(r \mid t, \text{Iters})$, weighting function $w(t)$.
**Init:** $\theta \leftarrow \phi$, $\text{Iters} = 0$.
**repeat**
    **if** ECD **then**
        Sample $\mathbf{x}_0 \sim \mathcal{D}$
    **else if** Data-Free ECD **then**
        Sample $\boldsymbol{\epsilon}' \sim p(\boldsymbol{\epsilon})$
        Compute $\mathbf{x}_0 = f_{\text{sg}(\theta)}(\boldsymbol{\epsilon}', T)$                           ▷ sg is stop-gradient operator
    **end if**
    Sample $\boldsymbol{\epsilon} \sim p(\boldsymbol{\epsilon})$, $t \sim p(t)$, $r \sim p(r \mid t, \text{Iters})$
    Compute $\mathbf{x}_t = \mathbf{x}_0 + t \cdot \boldsymbol{\epsilon}$, $\Delta t = t - r$
    Compute $\mathbf{x}_r = \text{Solver}(\mathbf{x}_t, t \rightarrow r, f_\phi)$
    $L(\theta) = w(t) \cdot \mathbf{d}(f_\theta(\mathbf{x}_t), f_{\text{sg}(\theta)}(\mathbf{x}_r))$
    $\theta \leftarrow \theta - \eta \nabla_\theta L(\theta)$
    $\text{Iters} = \text{Iters} + 1$
**until** $\Delta t \rightarrow 0$ **return** $\theta$                                                        ▷ ECD

---

## D EXPERIMENTAL DETAILS

**Model Setup.** For both unconditional and class-conditional CIFAR-10 experiments, we initial ECMs from the pretrained EDM (Karras et al., 2022) of DDPM++ architecture (Song et al., 2021b). For class-conditional ImageNet 64×64 experiments, we initial ECM-S/M/L/XL, ranging from 280M to 1.1B, from the pretrained EDM2 (Karras et al., 2024). Detailed model configurations are presented in Tab. 8.

We follow (Karras et al., 2022; Song et al., 2023) and set $c_{\text{skip}}(t) = \sigma_{\text{data}}^2/(t^2 + \sigma_{\text{data}}^2)$ and $c_{\text{out}}(t) = t\sigma_{\text{data}}/\sqrt{t^2 + \sigma_{\text{data}}^2}$, where $\sigma_{\text{data}}^2$ is the variance of (normalized) data, and set to 0.5 for both CIFAR-10 and ImageNet 64×64.

**Computational Cost.** ECT is computationally efficient. On ImageNet 64×64, the tuning stage of ECT requiring only 0.39% of the iCT (Song and Dhariwal, 2023) training budget, and 0.60% to 1.91% of the EDM2 (Karras et al., 2024) pretraining budget depending on the model sizes. The exact computational resources required to train each individual model are shown in Tab. 8.

**Training Details.** We use RAdam (Liu et al., 2019) optimizer for experiments on CIFAR-10 and Adam (Kingma and Ba, 2014) optimizer for experiments on ImageNet 64×64. We set the $\beta$ to (0.9, 0.999) for CIFAR-10 and (0.9, 0.99) for ImageNet 64×64. All the hyperparameters are indicated in

Table 8: Model Configurations and Training Details for unconditional and class-conditional ECMs on CIFAR-10, and ECM-S/M/L/XL on ImageNet 64×64.

| Model Setups | Uncond CIFAR-10 | Cls-Cond CIFAR-10 | ImageNet 64×64 | | | |
|---|---|---|---|---|---|---|
| | | | ECM-S | ECM-M | ECM-L | ECM-XL |
| Model Channels | 128 | 128 | 192 | 256 | 320 | 384 |
| Model capacity (Mparams) | 55.7 | 55.7 | 280.2 | 497.8 | 777.5 | 1119.3 |
| Model complexity (GFLOPs) | 21.3 | 21.3 | 101.9 | 180.8 | 282.2 | 405.9 |
| **Training Details** | | | | | | |
| Training Duration (Mimg) | 12.8 | 12.8 | 12.8 | 12.8 | 12.8 | 12.8 |
| Minibatch size | 128 | 128 | 128 | 128 | 128 | 128 |
| Iterations | 100k | 100k | 100k | 100k | 100k | 100k |
| Dropout probability | 20% | 20% | 40% | 50% | 50% | 50% |
| Dropout feature resolution | - | - | $\leq 16 \times 16$ | $\leq 16 \times 16$ | $\leq 16 \times 16$ | $\leq 16 \times 16$ |
| Optimizer | RAdam | RAdam | Adam | Adam | Adam | Adam |
| Learning rate max ($\alpha_{\text{ref}}$) | 0.0001 | 0.0001 | 0.0010 | 0.0009 | 0.0008 | 0.0007 |
| Learning rate decay ($t_{\text{ref}}$) | - | - | 2000 | 2000 | 2000 | 2000 |
| EMA beta | 0.9999 | 0.9999 | - | - | - | - |
| **Training Cost** | | | | | | |
| Number of GPUs | 1 | 1 | 4 | 8 | 8 | 8 |
| GPU types | A6000 | A6000 | H100 | H100 | H100 | H100 |
| Training time (hours) | 24 | 24 | 8.5 | 8.5 | 12 | 15 |
| **Generative Performance** | | | | | | |
| 1-step FID | 4.54 | 3.81 | 5.51 | 3.67 | 3.55 | 3.35 |
| 2-step FID | 2.20 | 2.02 | 3.18 | 2.35 | 2.14 | 1.96 |
| **ECT Details** | | | | | | |
| Regular Weighting ($\bar{w}(t)$) | $1/(t-r)$ | $1/(t-r)$ | $1/t^2 + 1/\sigma_{\text{data}}^2$ | $1/t^2 + 1/\sigma_{\text{data}}^2$ | $1/t^2 + 1/\sigma_{\text{data}}^2$ | $1/t^2 + 1/\sigma_{\text{data}}^2$ |
| Adaptive Weighting ($w(\Delta)$) | ✓ | ✓ | ✓ | ✓ | ✓ | ✓ |
| Adaptive Weighting Smoothing ($c$) | 0.0 | 0.0 | 0.06 | 0.06 | 0.06 | 0.06 |
| Noise distribution mean ($P_{\text{mean}}$) | $-1.1$ | $-1.1$ | $-0.8$ | $-0.8$ | $-0.8$ | $-0.8$ |
| Noise distribution std ($P_{\text{std}}$) | 2.0 | 2.0 | 1.6 | 1.6 | 1.6 | 1.6 |

Tab. 8. We do not use any learning rate decay, weight decay, or warmup on CIFAR-10. We follow EDM2 (Karras et al., 2024) to apply an inverse square root learning rate decay schedule on ImageNet 64×64.

On CIFAR-10, we employ the traditional Exponential Moving Average (EMA). To better understand the influence of the EMA rate, we track three Power function EMA (Karras et al., 2024) models on ImageNet 64×64, using EMA lengths of 0.01, 0.05, and 0.10. The multiple EMA models introduce no visible cost to the training speed. Considering our training budget is much smaller than the diffusion pretraining stage, we didn't perform Post-Hoc EMA search as in EDM2 (Karras et al., 2024).

Experiments for ECT are organized in a non-adversarial setup to better focus and understand CMs and avoid inflated FID (Stein et al., 2024). We conducted ECT using full parameter tuning in this work, even for models over 1B parameters. Investigating the potential of Parameter Efficient Fine Tuning (PEFT) (Hu et al., 2021) can further reduce the cost of ECT to democratize efficient generative models, which is left for future research.

We train multiple ECMs with different choices of batch sizes and training iterations. By default, ECT utilizes a batch size of 128 and 100k iterations, leading to a training budget of 12.8M on ImageNet 64×64. We have individually indicated other training budgets alongside the relevant experiments, wherever applicable.

**Sampling Details.** We apply stochastic sampling for 2-step generation. For 2-step sampling, we follow (Song and Dhariwal, 2023) and set the intermediate $t = 0.821$ for CIFAR-10, and $t = 1.526$ for ImageNet 64×64.

Intriguingly, these sampling schedules, originally developed for iCT, also perform well with our ECMs. This effectiveness across different CMs and training methods likely links to the inherent characteristics of the datasets and the forward process. Developing a scientific approach to determine optimal intermediate sampling schedules for CMs remains an open research problem.

**Evaluation Metrics.** For both CIFAR-10 and ImageNet 64×64, FID and FD$_{\text{DINOv2}}$ are computed using 50k images sampled from ECMs. As suggested by recent works (Stein et al., 2024; Karras et al., 2024), FD$_{\text{DINOv2}}$ aligns better with human evaluation. We use `dgm-eval`[1] to calculate FD$_{\text{DINOv2}}$ (Stein et al., 2024) to ensure align with previous practice. Performance results for prior methods in Tab. 1 are reported from previous works (Song et al., 2023; Song and Dhariwal, 2023; Karras et al., 2024).

**Visualization Setups.** Image samples in Fig. 3 are from class `bubble` (971), class `flamingo` (130), class `golden retriever` (207), class `space shuttle` (812), classs `Siberian husky` (250), classs `ice cream` (928), class `oscilloscope` (688), class `llama` (355), class `tiger shark` (3).

Each triplet (left-to-right) includes from 2-step samples from ECM-S trained with 12.8M images, ECM-S trained with 102.4M images, and ECM-XL trained with 102.4M images.

**1 GPU Hour Prototyping Settings.** This configuration uses a fixed $\Delta t \approx \mathrm{d}t$ by setting $q = 256$ in our mapping function (corresponding to $\frac{r}{t} \approx 0.99$) and an EMA rate of 0.9993 for the model parameters. Using these settings, we run 8000 gradient descent steps with a batch size of 128 on a single A100 40GB GPU.

**Scaling of Training Compute.** For the results on scaling laws for training compute on CIFAR-10 shown in Fig. 4 (Left), we train 6 class-conditional ECMs, each with varying batch size and number of training iterations. All ECMs in this experiment are initialized from the pretrained class-conditional EDM.

The minimal training compute at $2^0$ scale corresponds to a total budget of 12.8M training images. The largest training compute at $2^5$ scale utilizes a total budget of 409.6M training images, at $2\times$ EDM pretraining budget.

The first two points of $2^0$ and $2^1$ on Fig. 4 (Left) use a batch size of 128 for 100k and 200k iterations, respectively. The third point of $2^2$ corresponds to ECM trained with batch sizes of 256 for 200k iterations. The final three points of $2^3$, $2^4$, and $2^5$ correspond to ECM trained with a batch size of 512 for 200k, 400k, and 800k iterations, respectively, with the smoothing factor $c = 0.03$ enabled in the adaptive weighting $w(\Delta)$. We use $\bar{w}(t) = 1/t$ as the timestep weighting function to train all these models as this $\bar{w}(t)$ achieves good performance on FD$_{\text{DINOv2}}$.

**Scaling of Model Size and Model FLOPs.** We include details of model capacity as well as FLOPs in Tab. 8 to replicate this plot on ImageNet 64×64.

On ImageNet 64×64, we scale up the training budgets of ECM-S and ECM-XL from 12.8M (batch size of 128 and 100k iterations) to 102.4M (batch size of 1024 and 100k iterations). We empirically find that scaling the base learning rate by $\sqrt{n}$ works well when scaling the batch size by a factor of $n$ when using Adam (Kingma and Ba, 2014) optimizer.

## E  LIMITATIONS

One of the major limitations of ECT is that it requires a dataset to tune DMs to CMs. Recent works developed data-free approaches (Luo et al., 2024; Gu et al., 2023; Yin et al., 2023; Zhou et al., 2024a) for diffusion distillation. The distinction between ECT and data-free methods is that ECT learns the *consistency condition* on a given dataset through the self teacher, while data-free methods transfer knowledge from a frozen diffusion teacher. This feature of ECT can be a potential limitation since the training data of bespoke models are unavailable to the public. However, we hold an optimistic view on tuning CMs using datasets different from pretraining. Synthetic data, data composition, and data scaling for consistency models will be valuable research directions.

---

[1] https://github.com/layer6ai-labs/dgm-eval

# F QUALITATIVE RESULTS

We provide some randomly generated 2-step samples from ECMs trained on CIFAR-10 and ImageNet-$64 \times 64$ in Fig. 8 and Fig. 9, respectively.

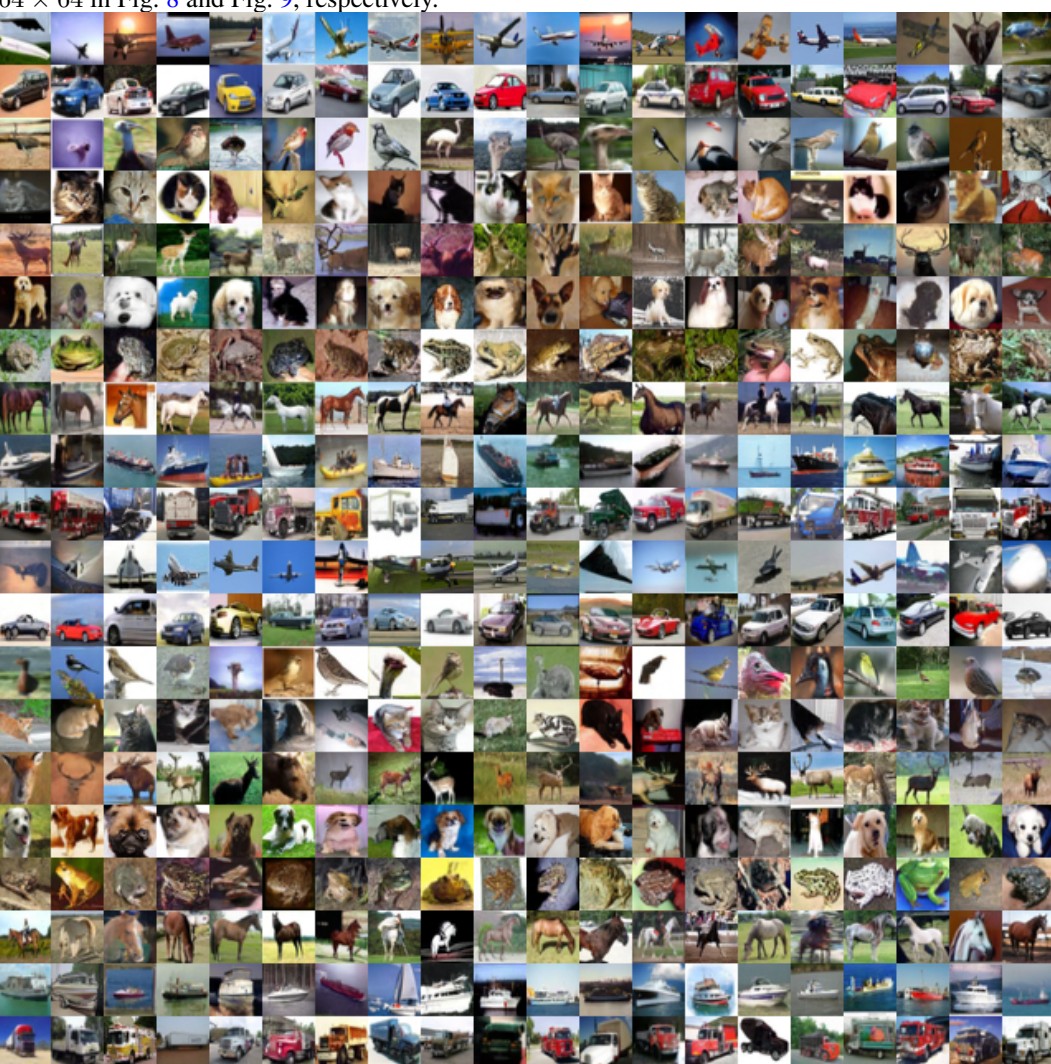

Figure 8: 2-step samples from class-conditional ECM trained on CIFAR-10. Each row corresponds to a different class.

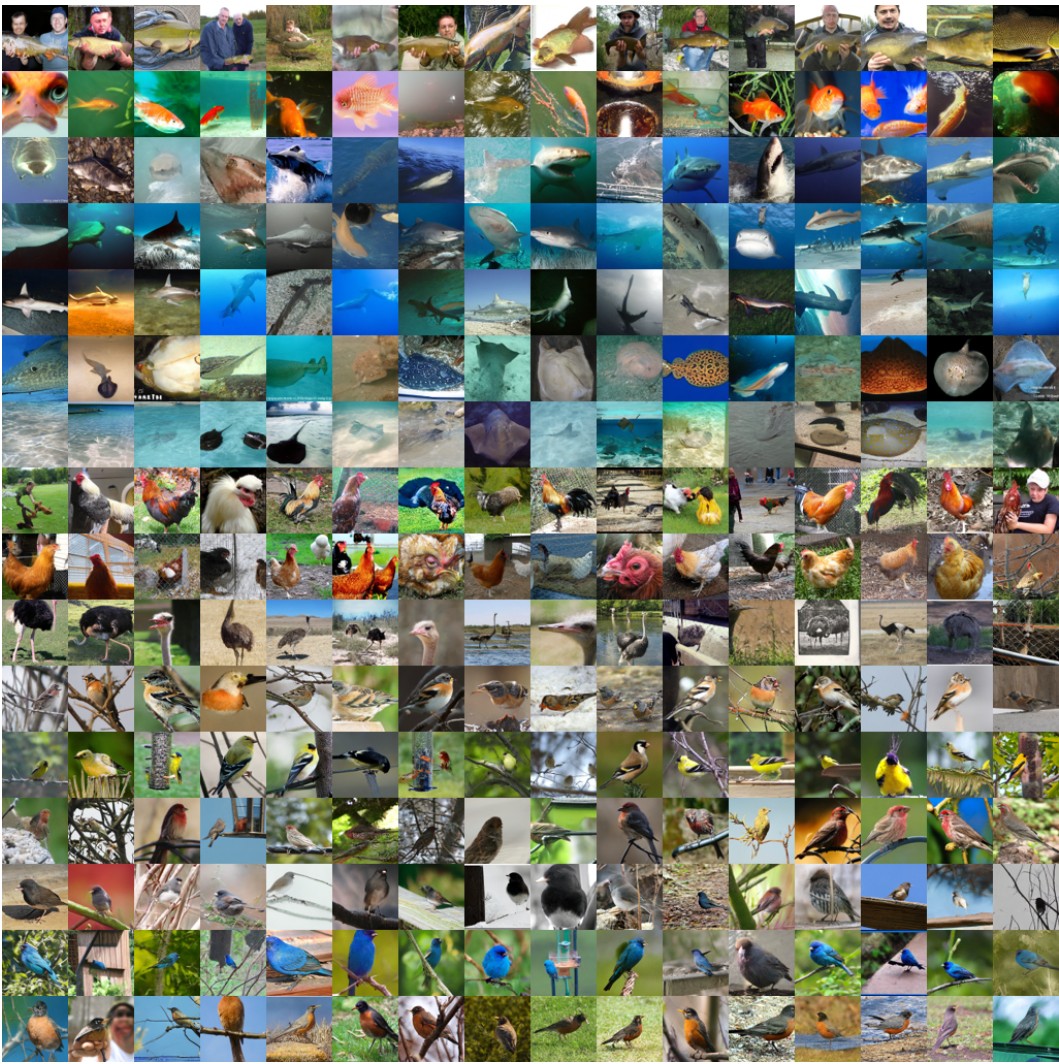

Figure 9: 2-step samples from class-conditional ECM-XL trained on ImageNet 64×64. Each row corresponds to a different class.

