# OpenReview forum: "Consistency Models Made Easy"
_ICLR.cc/2025/Conference — ICLR 2025 Poster_

### Official Review · Reviewer_EoPV · 2024-10-28

**Soundness:** 3
**Presentation:** 2
**Contribution:** 3
**Rating:** 8
**Confidence:** 4

**Summary:**

This paper describes a new recipe for consistency model training. By noticing that consistency training at high discretization levels amounts to the training of a diffusion model, the authors propose to initialize the consistency model with the weights of a pre-trained diffusion model. The actual consistency model training phase is then adapted to interpolate between diffusion and continuous-time consistency training. Supplemented by additional changes to the metric and its weighting, this new recipe is shown to present favorable experimental properties: outperforming consistency distillation and training, as well as diffusion models, with a fraction of the training cost. This efficiency is highlighted in a study of the scaling laws of the proposed method.

**Strengths:**

The paper's strengths are immediate: by presenting a simple, easily actionable method to improve the performance of consistency models while significantly reducing their training cost, **the presented method has the potential to be used as a strong baseline for future research** in consistency models.

The paper is overall **well written**. The method is sufficiently motivated with the described insights on consistency training discretization, making diffusion pre-training an **organic improvement**. The **clarity** of its exposition, its available codebase -- that I could not assess in detail -- and, perhaps more importantly, its **simplicity** are important factors in its reusability.

The experimental results confirm the appeal of the new consistency training recipe with not only noticeable performance boosts, but also significant **efficiency gains**. The efficiency advantage is not only significant compared to standard consistency training, but even stronger compared to consistency distillation. Finally, even though no comparison with another method is provided, the presented scaling laws are an interesting addition.

**Weaknesses:**

While I believe this paper provides a valuable and actionable contribution, a few weaknesses, listed below in decreasing order of importance, prevent me from providing a positive recommendation.

## Clarification of the novelty of some contributions

The idea of **initializing consistency models with a pre-trained diffusion model has already been considered** in the initial paper of Song et al. (2023, Appendix B.3), albeit in a restricted setting and without the experimental value highlighted in the present paper. Still, this should be discussed and may have consequences on the empirical study (see next weakness).

Section 3.1 remains **vague whether the presented derivations are contributions of the present paper** or reformulated background from consistency models. My understanding is that these derivations, including the final loss function, are all already included in the original paper of Song et al. (2023): the differential equation appears in another form in their Appendix Remark 4, and the loss function closely resembles the original loss function (by removing scheduling information). This should be clarified.

**The choice of weighting function appears as a reformulation** of Song & Dhariwal (2023), unless I misunderstood its description in the paper. The insight of the pseudo-Huber metric providing an adaptive weighting term is interesting, but I do not see how the proposed alternative (using the $\ell_2$ as metric and modulate it with the weighting that the pseudo-Huber metric would have brought) is any different. Could the authors clarify this?

## Lacking experimental insights

While appealing, the presented experimental results lack additional insights and ablations to fully support the claims of the paper and highlight its added value w.r.t. prior work. Two main components are missing.

The first missing component is a **full fledged ablation study**. One is already included in Appendix Section B (Table 5). However, given this preliminary result showing that the simple diffusion pre-training (already considered in prior work, cf. previous weakness) brings most of the improvements, I think the paper would benefit from including in its main part an augmented ablation study in the same experimental setting as Section 4. This ablation study should then be discussed to assess the significance of each contribution in the new training recipe.

The second missing component is a **study of the evolution of the methods' performance w.r.t. training time/iterations**. As is, baseline and ablation results are presented at a fixed number of iterations. Plotting their performance w.r.t. the number of iterations would enable a more comprehensive efficiency comparison, as well as justify the choice of stopping time for baselines in e.g. Table 1.

## Lack of moderation and rigor in some assertions

This is a less important issue that still deserves to be addressed. The following statements lack moderation and/or rigor.
- In Section 1, it is stated that "the speedup achieved by these sampling techniques usually comes at the expense of the quality of generated samples". This is partially incorrect as some works like the one of Karras et al. (2022) significantly reduced the number of required model evaluations while maintaining performance.
- It is incorrect that the proposed loss function of Eq. (12) "generalizes Song et al. (2023)’s discrete training schedule to continuous time". The proposed loss indeed relies on discretizing the aforementioned differential equation. Instead, this loss is exactly the consistency training loss of Song et al. (2023) untied from its specific discretization grid.
- Writing $\Delta t \to \mathrm{d}t$ instead of $0$ is confusing.
- In Section 4.1, the proposed model is said to "only [require] 1/1000 of [Score SDE's] inference cost". This is correct but misleading as state-of-the-art diffusion models no longer require thousands of model evaluations. This statement should be toned down.

## Minor issues

- The diffusion ODE of Eq. (2), from Karras et al. (2022), requires that $\mathbf{f} = 0$ in Eq. (1).
- $f(\mathbf{x}_t, t)$ is said to be "a denoising function" after Eq. (3). This statement should be more detailed.
- To my knowledge, in Karras et al. (2022)'s timestep schedule, $\rho = 7$ instead of $0.7$ on line 134.
- To my understanding, a part of the numbers in Table 1 were not obtained by the authors but are reported from other papers. This should be clarified.
- $\epsilon$ in Eq. (19) should be $c^2$ to be consistent with the rest of the paper.

**Questions:**

Given the above weaknesses but also the potential impact of its contribution, I think that this paper barely misses the acceptance threshold of ICLR, hence my recommendation of "5: marginally below the acceptance threshold". My questions are detailed in the "Weaknesses" section. I am willing to update my score following the authors' response as I believe most of them can be answered with a revision of the paper during the discussion period.

I would have two additional questions.
- Consistency distillation is tested following the original setup of Song et al. (2023). How would it benefit from later improvements (even though presented only in the setting of consistency training) of iCT (Song & Dhariwal, 2023)?
- I would be interested in hearing the authors' opinion on whether the presented model is a distillation model or not. This is not discussed explicitly in the paper.

----
### Post-rebuttal update

The authors have addressed many of my concerns, so I choose to now recommend an "accept". **Nonetheless**, in case the paper is accepted, **I would like the authors to further improve the paper as suggested in the thread below**.

---

> ### Comment · Reviewer_EoPV · 2024-11-27
>
> I would like to thank the authors for their comprehensive response. Given that many of my concerns have been addressed (cf. details below), and considering the high potential impact of the presented method (due to its simplicity and significant performance improvements both in the original submission and in the rebuttal), I believe this work should be accepted. Hence, I choose to raise my score to an "accept".
>
> **Nonetheless**, in case the paper is accepted, **I would like the authors to further improve the paper as suggested below**.
>
> I look forward to discussing with the other reviewers about the submission.
>
> ### Novelty
>
> I am satisfied with the changes in the writing about their contributions. Still, I agree with Reviewer LAzX that **Section 3.1 could be moved to Section 2 for the sake of clarity of exposition**.
>
> I now better understand the paper's contributions on weighting and thank the authors for clarifying my confusion.
>
> ### Experimental insights
>
> I acknowledge the ablation study. I think **both tables would deserve to be included in the main paper and in the same experimental setting as the main experiments**. In the current state of the paper, the ablations are quite disconnected and it hinders a comprehensive overview of the empirical assessment of the contributions.
>
> I appreciate the added **Figure 7 (right)** on the evolution of FID during training, but **it should include at least one baseline** (typically iCT) to provide a proper efficiency comparison.
>
> ### Moderation and rigor + Minor issues
>
> I thank the authors for taking into account my suggestions.
>
> ### New experiments
>
> During the discussion period, the authors provided several additional results (e.g. ImageNet 512, new demonstration of the utility of their weighting mechanism). **Can the authors make sure to include them in the paper?** I am not sure they were added.
>
> ### Distillation vs tuning
>
> I would like to thank the authors for their response and additional experiment on consistency distillation. This question naturally arises because the presented method also requires pre-training a diffusion model. Therefore, I think **this discussion and this experiment should be included in the paper** to refine the related work section and further highlight the benefits of the new training recipe.

---

> > ### Author Response · Authors · 2024-11-28
> >
> > Dear Reviewer,
> >
> > Thank you for the feedback. We greatly appreciate your recognition of the **potential impact** of our work.
> >
> > Per your suggestion, we have refactored Section 2 to incorporate the content previously in Section 3.1 and clarified the distinction between ECT as a distillation or training method.
> >
> > Following your recommendations and the discussions during the rebuttal, we will further improve the manuscript by refining the ablation studies and training evolution and including experiments and demonstrations (ImageNet 512×512, weighting schemes, and distillation results). We hope to provide a more comprehensive presentation of our method through further revision.
> >
> > Thank you again for the constructive discussions and for improving this work together.
> >
> > Best regards,
> >
> > Authors

---

### Official Review · Reviewer_QQZ4 · 2024-10-29

**Soundness:** 3
**Presentation:** 2
**Contribution:** 2
**Rating:** 5
**Confidence:** 4

**Summary:**

This paper aims to improve the training efficiency of CMs by proposing three techniques: "continuous-time schedule," "dropout," and "weighting function." The resulting method demonstrates good performance. Additionally, the paper explores a phenomenon called the "curse of consistency" as well as the scaling laws of ECT.

**Strengths:**

1. This paper explores the concept of the "curse of consistency," which presents an intriguing perspective.
2. The method proposed in this paper achieves good performance.
3. This paper discusses the “scaling laws”.

**Weaknesses:**

1. While the "curse of consistency" is indeed fascinating, discussing only the upper bound fails to capture the true nature of errors. An increase in the upper bound does not necessarily indicate a corresponding increase in error.
2. Since the primary advantage of your method lies in its training speed, I believe you may have overlooked an important scenario: "pretraining + iCT tuning," as illustrated in Figure 2.
3. The relationship between your primary observation and your main method is not clear. Specifically, I find it difficult to understand the necessity of proposing a "continuous-time training schedule" and a "weighting function" to address the "curse of consistency" problem. If none of your techniques are linked to the "curse of consistency," what is the rationale for mentioning it, and how does this discussion relate to the overall objectives of your paper? In fact, the "curse of consistency" is not even addressed in your abstract.
4. You should place the ablation studies of your techniques (Table 6) in your main paper instead of the appendix.
5. The "dropout" technique appears to be quite significant; could you clarify why it is placed in the appendix?

**Questions:**

I think all of my questions are presented in the “weaknesses” part. If you can solve most of them well, I will raise my score.

---

### Official Review · Reviewer_LAzX · 2024-10-30

**Soundness:** 3
**Presentation:** 3
**Contribution:** 3
**Rating:** 8
**Confidence:** 4

**Summary:**

The authors propose easy consistency tuning (ECT), a method to train consistency models (CMs) using less compute than competing methods by initializing them from pre-trained diffusion models (DMs) and then fine-tuning them for few-step generation. The fine-tuning step enables faster convergence compared to training CMs from scratch (consistency training, CT) and avoids accumulating errors from a frozen teacher DM compared to consistency distillation (CD). Training CMs with ECT achieves state-of-the-art performance on both CIFAR-10 and ImageNet 64×64. The authors additionally explore scaling laws in the fine-tuning step, demonstrating performance gains with increased training compute.

**Strengths:**

**Results.**
The primary strength of this work lies in its empirical results, achieving state-of-the-art performance on standard image generation benchmarks. Overall, the experimental analysis is quite comprehensive by comparing against (and outperforming) recent and strong baseline methods in Table 1, and ablating some key design choices in the appendix. The ECT scaling laws in Section 4.2 are an interesting and underexplored direction in CMs, and the authors provide evidence of improved sample quality with increased training compute. If this were to also hold true for both higher dimensional datasets (e.g., ImageNet 512x512, LAION) and latent DMs (e.g., Stable Diffusion), it would have important implications for few-step generation at scale.

**Insights.**
Furthermore, some of the insights provided in the methodology of this paper are quite interesting. Specifically, Section 3.2 provides some interesting discussion on the limitations of training CMs from scratch, dubbed the “curse of CMs”, by bounding the single-step prediction error as a function of the granularity of discretization.

**Weaknesses:**

**Novelty.**
Despite the strong results demonstrated in this work, I have questions about its novelty. The main methodological contribution is to initialize CM training (CT) with a pre-trained DM to enable faster convergence. In the setting of consistency distillation (CD) and DM distillation in general, it is already common practice to initialize the student from the weights of the teacher DM, effectively reducing distillation to a fine-tuning task to reduce computational requirements and facilitate convergence, similar to what’s being motivated here. The difference in this work compared to CD is to instead initialize a CM from a DM in the setting of CT, which alleviates DM error accumulation as in the case of CD. However, this seems like a rather simple extension of CMs and the connection between CMs and DMs does not seem particularly novel, so I would ask the authors to clarify their proposed novelty here.

Moreover, the authors formulate CMs in continuous-time which was already done in the seminal CM work by Song et al. in Appendix B.2, so it’s not clear what the difference is here if any. Along this vein, I would also ask the authors to clarify the purpose of Section 3.1 and why it’s necessary to formulate CMs in terms of the differential condition since it currently reads as a seemingly disconnected derivation and it’s not clear from reading the paper why this formulation is needed for the rest of their method instead of using the standard discrete/continuous CM formulation by Song et al. In addition, the particular weighting function derived in Section 3.1, $w(t,r)=\frac{1}{t-r}$, does not seem to be what the authors actually end up using in practice since, in Section 3.3, the authors go on to say “Instead, we consider decoupled weighting functions without relying on $p(r|t)$” in L311-L312. Can the authors please clarify these points and also update the manuscript accordingly.

**Ablations.**
Given that the central contribution of ECT is initializing CMs from a DM, it would be important to see the performance of training ECT from scratch (i.e., w/o initializing from a DM) while keeping all other ECT design choices fixed, namely those outlined in Section 3.3 in terms of training schedule, loss function, and weighting function. Can the authors please provide this result in their rebuttal as it is a relevant ablation?

**Easiness.**
The authors claim that their method makes CMs easy to train but, from a design perspective, various choices (e.g., Equation 15) do not necessarily seem more intuitive or “easier” compared to those in the original CT or in the follow-up iCT. I think it would be fair to say that ECT makes training CMs more accessible and more computationally efficient but it’s not clear that “easy” is the right characterization. Can the authors please clarify their position?

**Questions:**

In addition to my questions already mentioned in the paper’s weaknesses, I have a few other minor questions and clarifications:
- I do not count this as a weakness since CIFAR-10 and ImageNet 64×64 are standard benchmarks, but have the authors also considered evaluating their method on higher dimensional datasets (e.g., ImageNet 512x512, LAION) and/or latent diffusion models (e.g., Stable Diffusion)? It would certainly be interesting and further strengthen results to position ECT as the de facto standard training regime for CMs more generally.
- Why do the authors use FD-DINOv2 for Figure 4 (left) and use FID for Figure 4 (right)? It seems inconsistent to use different metrics here.
- In Figure 2, why is the diffusion pre-training compute for both CD and ECT less than that for SDE/DM? Are the authors using a different DM?

Overall, should the authors adequately address my concerns and questions, I will consider increasing my score.

---
As per my comment below:

Thank you to the authors for continuing to engage in discussion and for acting in good faith throughout this rebuttal process by incorporating my feedback into the updated manuscript. The novelty point has been clarified, and the improved separation of pedagogical and methodological content between Sections 2 and 3 helps better focus Section 3 on the paper's contributions. I believe the authors have addressed most of my concerns/questions, so I will increase my score to 8. With the method's simplicity, combined with its strong empirical results, my hope is that the community will build on this framework for training consistency models.

---

### Official Review · Reviewer_FML7 · 2024-11-03

**Soundness:** 2
**Presentation:** 4
**Contribution:** 2
**Rating:** 6
**Confidence:** 4

**Summary:**

In this paper, the athors made the point that diffusion models can be viewed as a special case of CMs. Based on it, they fine-tuned a consistency model starting from a pretrained diffusion model and progressively approximate the full consistency condition to stronger
degrees over the training process. The experiments verified its efficiency.

**Strengths:**

The approach is very efficent as they showed. It could be used to greatly mprove efficiency and
performance of CMs at a large scale.

**Weaknesses:**

The main motivation of the paper is straightforward. It is hard for the reader to fullly trust their obverstaion that  diffusion models can be viewed as a special case of CMs in practice. The data sets and metircs on the images generation are limited. More extensive experiments or analysis should be conducted to justify their claims.

**Questions:**

1. Pseudo-Huber metric is adopted in the model. Is there any other alternative?
2. In the experiments, it is better to provide more metrics other than FID only.

---

### Meta-Review · Area_Chair_GNqZ · 2024-12-12

**Metareview:**

This paper explores a method to train consistency models by initializing them from pre-trained diffusion models and then fine-tuning. Various tricks are employed in the training to improve convergence and efficiency of the training, resulting in SOTA performance on small-scale image generation baselines with low cost. Scaling laws are also discussed.

Reviewers were initially critical of some of the presentation choices and felt that prior work had not been sufficiently acknowledged. They wanted the novelty of the proposed approach clarified in comparison to past work. Reviewers were also concerned about the lack of experimental insights and ablations, along with several other more minor issues. Largely these concerns were addressed in the rebuttal phase, with better referencing and reorganization of the paper’s structure, as well as new experiments that filled in existing gaps in the narrative.

I am recommending acceptance as a Poster, but ask the authors to ensure that the reviewer’s concerns on the writing is taken into account for the final version (especially the results of discussion with LAzX and EoPV on clarifying prior work).

**Additional Comments On Reviewer Discussion:**

The main points of concern raised by reviewers were: novelty and proper referencing of prior work; lack of experimental insights and ablations. There were several other more minor points of concern.

Generally, the authors were able to address these concerns, and have made progress on integrating the reviewer’s recommendations into their paper. With these changes, I agree with the majority of reviewers that the paper is scientifically sound and ready for ICLR.

---

### Decision · Program_Chairs · 2025-01-22

Accept (Poster)